# Size Lowerbounds for Deep Operator Networks

**Anirbit Mukherjee** *anirbit.mukherjee@manchester.ac.uk*
*Department of Computer Science*
*The University of Manchester*

**Amartya Roy**[*] *Amartya.Roy@in.bosch.com*
*Robert Bosch GmbH, Coimbatore, India*

**Reviewed on OpenReview:** *https://openreview.net/forum?id=RwmWODTNFE*

## Abstract

Deep Operator Networks are an increasingly popular paradigm for solving regression in infinite dimensions and hence solve families of PDEs in one shot. In this work, we aim to establish a first-of-its-kind data-dependent lowerbound on the size of DeepONets required for them to be able to reduce empirical error on noisy data. In particular, we show that for low training errors to be obtained on $n$ data points it is necessary that the common output dimension of the branch and the trunk net be scaling as $\Omega\left(\sqrt[4]{n}\right)$.

This inspires our experiments with DeepONets solving the advection-diffusion-reaction PDE, where we demonstrate the possibility that at a fixed model size, to leverage increase in this common output dimension and get monotonic lowering of training error, the size of the training data might necessarily need to scale at least quadratically with it.

## 1 Introduction

Data-driven approaches to analyze, model, and optimize complex physical systems are becoming more popular as Machine Learning (ML) methodologies are gaining prominence. Dynamic behaviour of such systems is frequently characterized using systems of Partial Differential Equations (PDEs). A large body of literature exists for using analytical or computational techniques to solve these equations under a variety of situations, such as various domain geometries, input parameters, and initial and boundary conditions. Very often one wants to solve a "parametric" family of PDEs i.e have a mechanism of quickly obtaining new solutons to the PDE upon variation of some parameter in the PDE like say the viscosity in a fluid dynamics model. This is tantamount to obtaining a mapping between the space of possible parameters and the corresponding solutions to the PDE. The cost of doing this task with conventional tools such as finite element methods (Brenner & Carstensen, 2004) is enormous since distinct simulations must be run for each unique value of the parameter, be it domain geometry or some input or boundary value. Fortuitously, in recent times there have risen a host of ML methods under the umbrella of "operator learning" to achieve this with more attractive speed-accuracy trade-offs than conventional methods, (Ray et al., 2023)

As reviewed in (Ray et al., 2023), we recognize that operator learning is itself a part of the larger program of rapidly increasing interest, "physics informed machine learning" (Karniadakis et al., 2021). This program encompasses all the techniques that are being developed to utilize machine learning methods, in particular neural networks, for the numerical solution of dynamics of physical systems, often described as differential equations. Notable methodologies that fall under this ambit are, Physics Informed Neural Nets (Raissi & Karniadakis, 2018), DeepONet (Lu et al., 2019), Fourier Neural Operator (Li et al., 2020b), Wavelet Neural Operator (Tripura & Chakraborty, 2022), Convolutional Neural Operators (Raonic et al., 2023) etc.

Physics-Informed Neural Networks (PINNs) have emerged as a notable approach when there is one specific PDE of interest that needs to be solved. To the best of our knowledge some of the earliest proposals of

---

[*]A part of the work was done while the author was at the Jadavpur University

this were made in, (Dissanayake & Phan-Thien, 1994; Lagaris et al., 1998; 2000). The modern avatar of this idea and the naming of PINNs happened in (Raissi et al., 2019). This learning framework involves minimizing the residual of the underlying partial differential equation (PDE) within the class of neural networks. Notably, PINNs are by definition an unsupervised learning method and hence they can solve PDEs with no need for knowing any sample solutions. They have demonstrated significant efficacy and computational efficiency in approximating solutions to PDEs, as evidenced by (Raissi et al., 2018), (Lu et al., 2021), (Mao et al., 2020), (Pang et al., 2019), (Yang et al., 2021), (Jagtap & Karniadakis, 2021), (Jagtap et al., 2020), (Bai et al., 2021), A detailed review of this field can be seen at, (Cuomo et al., 2022).

As opposed to the question being solved by PINNs, Deep Operator Networks train a pair of nets in tandem to learn a (possibly nonlinear) operator mapping between infinite-dimensional Banach spaces - which de-facto then becomes a way to solve a family of parameteric PDEs in "one-shot". Its shallow version was proposed in (Chen & Chen, 1995b) and more recently its deeper versions were investigated in (Lu et al., 2019) and its theoretical foundations laid in (Lanthaler et al., 2022a).

Till date numerous variants of DeepONet models (Park et al., 2023), (Liu & Cai, 2021), (Hadorn, 2022), (Almeida et al., 2022), (Lin et al., 2022), (Xu et al., 2022), (Tan & Chen, 2022), (Zhang et al., 2022), (Goswami et al., 2022) have been proposed and this training process takes place offline within a predetermined input space. As a result, the inference phase is rapid because no additional training is needed as long as the new conditions fall within the input space that was used during training.

Other such neural operators like FNO (Li et al., 2020b), WNO[(Tripura & Chakraborty, 2022) enable efficient and accurate solutions to complex mathematical problems, opening up new possibilities for scientific computing and data-driven modeling. They have shown promise in various scientific and engineering applications including physics simulations (Choubineh et al., 2023), (Gopakumar et al., 2023), (Li et al., 2022b), (Lehmann et al., 2023), (Li et al., 2022a), image processing (Johnny et al., 2022), (Tripura et al., 2023), and weather-modelling (Kurth et al., 2022), (Pathak et al., 2022).

A deep mystery with neural nets is the effect of their size on their performance. On one hand, we know from various experiments as well as theory that the asymptotically wide nets are significantly weaker than actual neural nets and they have very different training dynamics than what is true for practically relevant nets. But, it is also known that there are specific ranges of overparametrization at which the neural net performs better than at any lower size. Modern learning architectures exploit this possibility and they are almost always designed with a large number of training parameters than the size of the training set. It seems to be surprisingly easy to find overparametrized architectures which generalize well. This contradicts the traditional understanding of the trade-off between approximation and generalization, which suggests that the generalization error initially decreases but then increases due to overfitting as the number of parameters increases (forming a U-shaped curve). However, recent research has revealed a puzzling non-monotonic dependency on model size of the generalization error at the empirical risk minimum of neural networks. This curious pattern is referred to as the "double-descent" curve,(Belkin et al., 2019). Some of the current authors had pointed out (Gopalani & Mukherjee, 2021), that the nature of this double-descent curve might be milder (and hence the classical region exists for much large range of model sizes) for DeepONets - which is the focus of this current study.

It is worth noting that this phenomenon has been observed in decision trees and random features and in various kinds of deep neural networks such as ResNets, CNNs, and Transformers (Nakkiran et al., 2021). Also, various theoretical approaches have been suggested towards deriving the double-descent risk curve, (Belkin et al., 2018a), (Belkin et al., 2018b), (Deng et al., 2022), (Kini & Thrampoulidis, 2020).

In recent times, many kinds of generalization bounds for neural nets have also been derived, like those based on Rademacher complexity (Sellke, 2023), (Golowich et al., 2018), (Bartlett et al., 2017) which are uniform convergence bounds independent of the trained predictor or results as in (Li et al., 2020a) and (Muthukumar & Sulam, 2023) which have developed data-dependent non-uniform bounds. These help explain how the generalization error of deep neural nets might not explicitly scale with the size of the nets. Some of the current authors had previously shown (Gopalani et al., 2022) the first-of-its-kind Rademacher complexity bounds for DeepONets which does not explicitly scale with the width (and hence the number of trainable

parameters) of the nets involved. Despite all these efforts, to the best of our knowledge, it has generally remained unclear as to how one might explain the necessity for overparameterization for good performance in any such neural system.

In light of this, a key advancement was made in, (Bubeck & Sellke, 2023). They showed, that with high probability over sampling $n$ training data in $d$ dimensions, if there has to exist a neural net $f$ of depth $D$ and $p$ parameters such that it has empirical squared-loss error below a measure of the noise in the labels then it must be true that, $\mathrm{Lip}(f) \geq \tilde{\Omega}\left(\sqrt{\frac{nd}{Dp}}\right)$. This can be interpreted as an indicator of why large models might be necessary to get low training error on real world data. Building on this work, we prove the following result (stated informally) for the specific instance of operator learning as we consider,

**Theorem 1.1** (Informal Statement of Theorem 4.2)**.** *Suppose one considers a DeepONet function class at a fixed bound on the weights and the total number of parameters and both the branch and the trunk nets ending in a layer of sigmoid gates. Then with high probability over sampling a $n$−sized training data set, if this class has to have a predictor which can achieve empirical training error below a label noise dependent threshold, then necessarily the common output dimension of the branch and the trunk must be lower bounded as $\Omega\left(\sqrt[4]{n}\right)$. And notably, the prefactors suppressed by $\Omega$ scale inversely with the bound on the weights and the size of the model.*

Thus, to the best of our knowledge, our result here makes a first-of-its-kind progress with explaining the size requirement for DeepONets and in particular how that is related to the available size of the training data. Further, motivated by the above, we shall give experiments to demonstrate that at a fixed model size, for DeepONets to leverage an increase in the size of the common output dimension of branch and trunk, the size of the training data might need to be scaled at least quadratically with that.

The proof in (Bubeck & Sellke, 2023) critically uses the Lipschitzness condition of the predictors to leverage isoperimetry of the data distribution. And that is a fundamental mismatch with the setup of operator learning - since DeepONets are not Lipschitz functions. Thus our work embarks on a program to look for an analogous insight as in (Bubeck & Sellke, 2023) that applies to DeepONets.

## 1.1 The Formal Setup of DeepONets

We recall the formal setup of DeepONets (Ryck & Mishra, 2022). Given $T > 0$ and $D \subset \mathbb{R}^d$ compact, consider functions $u : [0, T] \times D \to \mathbb{R}^k$, for $k \geq 1$, that solve the following time-dependent PDE,

$$\mathcal{L}_a(u)(t, x) = 0 \quad \text{and} \quad u(x, 0) = u_0 \quad \forall (t, x) \in [0, T] \times D .$$

This abstracts out the use case of wanting to find the time evolution of $k$ dimensional vector fields on $d$ dimensional space and them being governed by a specific P.D.E. Further, let $\mathcal{H}$ be the function space of PDE solutions of the above. Define a function space $\mathcal{Y}$ s.t $u_0 \in \mathcal{Y} \subset L^2(D)$ i.e the space of initial conditions and then the differential operator above can be chosen to map as, $\mathcal{L}_a : \mathcal{H} \to L^2([0, T] \times D)$ and these operators are indexed by a function $a \in \mathcal{Z} \subset L^2(D)$.

Corresponding to the above we have the solution operator $\mathcal{G} : \mathcal{X} \to \mathcal{H} : f \mapsto u$, where $f \in \{u_0, a\}$ $\mathcal{X} \in \{\mathcal{Y}, \mathcal{Z}\}$ – where the two choices correspond to the two natural settings one might consider, that of wanting to solve the PDE for various initial conditions at a fixed differential operator or solve for a fixed initial condition at different parameter values for the differential operator.

The DeepONet architecture as shown in Figure 1 consists of two nets, namely a Branch Net and a Trunk Net. The former is denoted by $\mathcal{N}_{\mathrm{B}}$ that maps $\mathbb{R}^{d_1} \to \mathbb{R}^q$ - which in use will take as input a $d_1$ point discretization of a real valued function $f$ as a vector, $\mathbf{s} = (f(x_1), f(x_2), ..., f(x_{d_1}))$ corresponding to some arbitrary choice of "sensor points" $\{x_j \mid 1 \leq j \leq d_1\} \subset D$. The Trunk Net, denoted by $\mathcal{N}_{\mathrm{T}}$, maps $\mathbb{R}^{d_2} \to \mathbb{R}^q$ which takes as input any point in the domain of the functions in the solution space of the PDE. (In the context of the above PDE we would have $d_2 = 1 + d$). Note that in above $q$ is an arbitrary constant. Then the DeepONet with parameters $\theta$ (inclusive of both its nets) can be defined as the following map,

$$\mathcal{G}_\theta \left( \underbrace{f\left((x_1\right), f\left(x_2\right), \cdots, f\left(x_{d_1}\right))}_{\mathbf{s}} \right)(\mathbf{p}) \coloneqq \langle \mathcal{N}_{\mathrm{B}}(\boldsymbol{s}), \mathcal{N}_{\mathrm{T}}(\boldsymbol{p}) \rangle . \tag{1}$$

One would often want to constraint $\mathbf{p} \in U$ where $U$ compact domain in $\mathbb{R}^{d_2}$. Given the setup as described above, the objective of a DeepONet is to approximate the value $\mathcal{G}(f)(\mathbf{p})$ by $\mathcal{G}_\theta\left(f\left(x_1\right), f\left(x_2\right), \cdots, f\left(x_{d_1}\right)\right)(\mathbf{p})$.

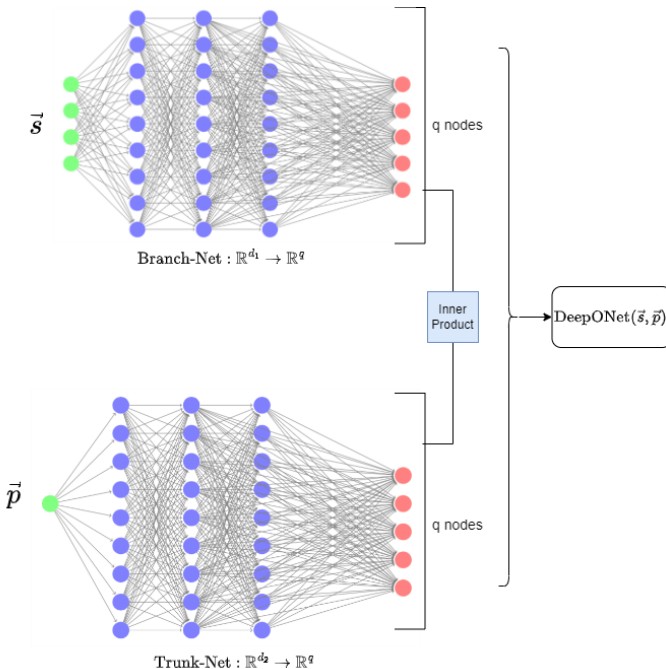

Figure 1: A Sketch of the DeepONet Architecture

**Review of the Universal Approximation Property of DeepONets** An universal approximation theorem for shallow DeepONets was established in (Chen & Chen, 1995a). A more general version of it was established in (Lanthaler et al., 2022b) which we shall now briefly review.

Consider two compact domains, $D \subset \mathbb{R}^d$ and $U \subset \mathbb{R}^n$, and two compact subsets of infinite dimensional Banach spaces, $K_1 \subset C(D)$ and $K_2 \subset C(U)$, where $C(D)$ represents the collection of all continuous functions defined on the domain $D$ and similarly for $C(U)$. We then define a (possibly nonlinear) continuous operator $\mathcal{G} : K_1 \to K_2$.

**Theorem 1.2.** (Restatement of a key result from (Lanthaler et al., 2022b) on Generalised Universal Approximation for Operators). *Let $\mu \in \mathcal{P}(C(D))$ be a probability measure on $C(D)$. Assume that the mapping $\mathcal{G} : C(D) \to L^2(U)$ is Borel measurable and satisfies $\mathcal{G} \in L^2(\mu)$. Then, for any positive value $\varepsilon$, there exists an operator $\tilde{\mathcal{G}} : C(D) \to L^2(U)$ (a DeepONet composed with a discretization map for functions in $C(D)$), such that*

$$\|\mathcal{G} - \tilde{\mathcal{G}}\|_{L^2(\mu)} = \left( \int_{C(D)} \|\mathcal{G}(u) - \tilde{\mathcal{G}}(u)\|_{L^2(U)}^2 d\mu(u) \right)^{1/2} < \varepsilon$$

In other words, $\tilde{\mathcal{G}}$ can approximate the original operator $\mathcal{G}$ arbitrarily close in the $L^2(\mu)$-norm with respect to the measure $\mu$. The above approximation guarantee between DeepONets and solution operators of differential equations ($\mathcal{G}$) clearly motivates the use of DeepONets for solving differential equations.

## 2 Related Works

In (Lanthaler et al., 2022b) the authors have defined the DeepONet approximation error as follows,

$$\widehat{\mathcal{E}} = \left( \int_{C(D)} \int_U |\mathcal{G}(u)(y) - \mathcal{N}(\boldsymbol{u})(y)|^2 \ \mathrm{d}y \ \mathrm{d}\mu(u) \right)^{1/2},$$

where the DeepONet approximates the underlying operator $\mathcal{G} : C(D) \to C(U)$ and $\mu$ being as defined previously and $\boldsymbol{u}$ a fixed finite grid discretization of $u$. To the best of our knowledge, the following is the only DeepONet size lowerbound proven previously,

**Theorem 2.1.** *Let $\mu \in \mathcal{P}\left(L^2(\mathbb{T})\right)$. Let $u \mapsto \mathcal{G}(u)$ denote the operator, mapping initial data $u(x)$ to the solution at time $t = \pi/2$, for the Burgers' PDE (Hon & Mao, 1998). Then there exists a universal constant $C > 0$ (depending only on $\mu$, but independent of the neural network architecture), such that the DeepONet approximation error $\widehat{\mathcal{E}}$ is,*

$$\widehat{\mathcal{E}} \geqslant \frac{C}{\sqrt{q}}$$

*where, $q$ is the common output dimension of the branch and the trunk net.*

*Firstly,* from above it does not seem possible to infer any relationship between the size of the neural architecture required for any specified level of performance and the amount of training data that is available to use. And that is a key connection that is being established in our work. *Secondly,* the above lowerbound is in the setting where there is no noise in the training labels - while our bound specifically targets to understand how the architecture is constrained when trying to get the empirical error to be below a measure of noise in the labels i.e our setup is that of solving PDEs in a supervised way while accounting for uncertainty in data. *Thirdly,* and most critically, the lower bound above is specific to Burger's PDE, while our theorem is PDE-independent.

**Organization** Starting in the next section we shall give the formal setup of our theory. In Section 4 we shall give the full statement of our theorem. In Section 5 we shall state all the intermediate lemmas that we need and their proofs. In Section 6 we give the proof of our main theorem. Motivated by the theoretical results, in Section 7 we give an experimental demonstration revealing a property of DeepONets about how much training data is required to leverage any increase in the common output dimension of the branch and the trunk. We conclude in Section 8 delineating some open questions.

## 3 Our Setup

In this section, we will give all the definitions about the training data and the function spaces that we shall need to state our main results. As a specific illustration of the definitions, we will also give an explicit example of a DeepONet loss function.

**Definition 1. Training Datasets**
*Let $(y_i, (\boldsymbol{s}_i, \boldsymbol{p}_i))$ be i.i.d. sampled input-output pairs from a distribution on $[-B, B] \times D \times U$ where $D$ and $U$ are compact subsets of $\mathbb{R}^{d_1}$ and $\mathbb{R}^{d_2}$ respectively and we define the conditional random variable $g(\boldsymbol{s}_i, \boldsymbol{p}_i) \coloneqq \mathbb{E}[y \mid (\boldsymbol{s}_i, \boldsymbol{p}_i)]$.*

**Definition 2. Branch Functions & Trunk Functions**

$$\mathcal{B} \coloneqq \{B_{\boldsymbol{w}} \ a \ function \ with \ \leq d_B \ parameters \mid B_{\boldsymbol{w}} : \mathbb{R}^{d_1} \to \mathbb{R}^q, \ \mathrm{Lip}(B_{\boldsymbol{w}}) \leq L_B \ \& \ \|\boldsymbol{w}\|_2 \leq W_B \ \& \ \|B_{\boldsymbol{w}}\|_\infty \leq \mathcal{C}\}$$

$$\mathcal{T} \coloneqq \{T_{\boldsymbol{w}} \ a \ function \ with \ \leq d_T \ parameters \mid T_{\boldsymbol{w}} : \mathbb{R}^{d_2} \to \mathbb{R}^q, \ \mathrm{Lip}(T_{\boldsymbol{w}}) \leq L_T \ \& \ \|\boldsymbol{w}\|_2 \leq W_T \ \& \ \|T_{\boldsymbol{w}}\|_\infty \leq \mathcal{C}\}$$

*The functions in the set $\mathcal{B}$ shall be called the "Branch Functions" and the functions in the set $\mathcal{T}$ would be called the "Trunk Functions".*

The bound of $\mathcal{C}$ in the above definitions abstracts out the model of the branch and the trunk functions being nets having a layer of bounded activation functions in their output layer - while they can have any other activation (like ReLU) in the previous layers.

**Definition 3. DeepONets**
*Given the function classes in Definition 2, we define the corresponding class of DeepONets as,*

$$\mathcal{H} \coloneqq \{h_{\boldsymbol{w}_b, \boldsymbol{w}_t} = h_{(\boldsymbol{w}_b, \boldsymbol{w}_t)} \mid \mathbb{R}^{d_1} \times \mathbb{R}^{d_2} \ni (\boldsymbol{s}, \boldsymbol{p}) \mapsto \boldsymbol{h}_{\boldsymbol{w}_b, \boldsymbol{w}_t}(\boldsymbol{s}, \boldsymbol{p}) \coloneqq \langle B_{\boldsymbol{w}_b}(\boldsymbol{s}), \mathbf{T}_{\boldsymbol{w}_t}(\boldsymbol{p}) \rangle \in \mathbb{R}, B_{\boldsymbol{w}_b} \in \mathcal{B} \ \& \ \mathbf{T}_{\boldsymbol{w}_t} \in \mathcal{T}\}$$

*Further, note that $\forall \theta > 0 \ \exists$ a "$\theta$-cover" of this function space $\mathcal{H}_\theta$ such that, $\forall h_{\boldsymbol{w}_b, \boldsymbol{w}_t} \in \mathcal{H}, \exists h_{(\boldsymbol{w}_{b, \frac{\theta}{2}}, \boldsymbol{w}_{t, \frac{\theta}{2}})} \in \mathcal{H}_\theta$ s.t $\left\| \boldsymbol{w}_b - \boldsymbol{w}_{b, \frac{\theta}{2}} \right\| \leq \frac{\theta}{2}$ and $\left\| \boldsymbol{w}_t - \boldsymbol{w}_{t, \frac{\theta}{2}} \right\| \leq \frac{\theta}{2}$ and $\boldsymbol{w}_{b, \frac{\theta}{2}}$ and $\boldsymbol{w}_{t, \frac{\theta}{2}}$ being elements of the $\frac{\theta}{2}$ covering space of the set of branch and trunk weights respectively.*

It's easy to see how the above definition of $\mathcal{H}$ includes functions representable by the architecture given in Figure 1. Now we recall the following result about neural nets from (Bubeck & Sellke, 2023).

**Lemma 3.1.** *Let $f_{\boldsymbol{w}}$ be a neural network of depth $D$, mapping into $\mathbb{R}$ with the vector of parameters being $\boldsymbol{w} \in \mathbb{R}^p$ and all the parameters being bounded in magnitude by $W$ i.e the set of neural networks parametrized by $\boldsymbol{w} \in [-W, W]^p$. Let $Q$ be the maximum number of matrix or bias terms that are tied to a single parameter $w_a$ for some $a \in [p]$. Corresponding to it we define, $B(w) \coloneqq \prod_{j \in [D]} \max\left(\left\|\boldsymbol{W}_j\right\|_{op}, 1\right)$, where $\boldsymbol{W}_j$ is the matrix in the $j^{th}$-layer of the net.*

*Let $\boldsymbol{x} \in \mathbb{R}^d$ such that $\|\boldsymbol{x}\| \leq R$, and $\boldsymbol{w}_1, \boldsymbol{w}_2 \in \mathbb{R}^p$ such that $B(\boldsymbol{w}_1), B(\boldsymbol{w}_2) \leq \bar{B}$. Then one has*

$$|f_{\boldsymbol{w}_1}(\boldsymbol{x}) - f_{\boldsymbol{w}_2}(\boldsymbol{x})| \leq \bar{B}^2 Q R \sqrt{p} \|\boldsymbol{w}_1 - \boldsymbol{w}_2\|.$$

*Moreover for any $\boldsymbol{w} \in [-W, W]^p$ with $W \geq 1$, one has, $B(\boldsymbol{w}) \leq (W\sqrt{pQ})^D$.*

In light of the above, we define $J$ as follows,

**Definition 4 (Defining $J$).** *Given any two valid weight vectors $\boldsymbol{w}_1$ and $\boldsymbol{w}_2$ for a "branch function" $B$ we assume to have the following inequality for some fixed $J > 0$,*

$$\sup_{\boldsymbol{s} \in D} \|B_{\boldsymbol{w}_1}(\boldsymbol{s}) - B_{\boldsymbol{w}_2}(\boldsymbol{s})\|_\infty \leq J \cdot \|\boldsymbol{w}_1 - \boldsymbol{w}_2\|$$

*And similarly for the trunk functions over the compact domain $U$.*

One can see that the above inequality is easy to satisfy if the space of inputs to the branch or the trunk is bounded. Thus invocation of this inequality implicitly constraints the support of the data distribution.

**An Example of a DeepONet Loss** To put the above definitions in context, let us consider an explicit example of using a DeepONet to solve the forced pendulum PDE, $\mathbb{R}^2 \ni \frac{\mathrm{d}(y,v)}{\mathrm{d}t} = (v, -k \cdot \sin(y) + f(t)) \in \mathbb{R}^2$ at different forcings $f$ at a given initial condition. Corresponding to this, the training data a DeepONet would need would be 3–tuples of the form, $(\boldsymbol{x}_B(f), x_T, y)$, where $\boldsymbol{x}_B(f)$ is a discretization of a forcing function $f$ onto a grid of "sensor points", $y \in \mathbb{R}$ is the angular position of the pendulum at time $t = x_T$ for $f$. Typically $y$ is a standard O.D.E. solver's approximate solution. It's clear that here $y$ being an angle is bounded, as was the setup in Definition 1.

Referring to Figure 1, we note that $\vec{s} = \boldsymbol{x}_B(f)$, a $d_1$ point discretization of a forcing function $f$ would be the input to the branch net, $\vec{p} = x_T \in \mathbb{R}^{d_2} = \mathbb{R}$, would be the trunk input i.e the time instant where we have the location ($y$) of the pendulum. Then the output of the architecture in Figure 1 is the evaluation of the following inner-product, $\mathbb{R}^{d_1} \times \mathbb{R}^{d_2} \ni (\vec{s}, \vec{p}) \mapsto \mathrm{DeepONet}(\vec{s}, \vec{p}) \coloneqq \langle \mathrm{Branch-Net}(\vec{s}), \mathrm{Trunk-Net}(\vec{p}) \rangle$. And given $n$ such data as above, the $\ell_2$ empirical loss would be, $\mathcal{L} \coloneqq \frac{1}{2n} \sum_{i=1}^{n} (y_i - \mathrm{DeepONet}(\boldsymbol{x}_B(f_i), x_{T,i}))^2$. This

empirical loss when minimized would yield a trained architecture of form as in Figure 1, which when queried for new forcing functions and time instants would yield accurate estimates of the corresponding pendulum locations.

## 4 The Main Theorem

In the setup of the definitions given above, now we can state our main result as follows,

**Theorem 4.1.** $\forall \delta \in (0,1)$ *and an arbitrary positive constant* $\epsilon > 0$ *and for* $\mathcal{C} \geq 1$ *(from Definition 2), if with probability at least* $1 - \delta$ *with respect to the sampling of the data* $\{(y_i, (\boldsymbol{s}_i, \boldsymbol{p}_i)) \mid i = 1, \ldots, n\}$, $\exists\, h_{\boldsymbol{w}_b, \boldsymbol{w}_t} \in \mathcal{H}$ *s.t*

$$\frac{1}{n} \sum_{i=1}^{n} (y_i - h_{\boldsymbol{w}_b, \boldsymbol{w}_t}(\boldsymbol{s}_i, \boldsymbol{p}_i))^2 \leq \sigma^2 - \epsilon \left(1 + \mathcal{C} \cdot J \cdot \left(B + 2 \cdot \mathcal{C}^2\right)\right)$$

*then,*

$$q \geq n^{\frac{1}{4}} \cdot \left( \frac{\epsilon^2}{288 \cdot B^2} \cdot \frac{1}{\ln\left(\left(\frac{4 \cdot \min\{d_B, d_T\}^2}{\epsilon}\right)^{d_B + d_T} \cdot \left(W_B \sqrt{d_B}\right)^{d_B} \cdot \left(W_T \sqrt{d_T}\right)^{d_T} + 2\right) + \ln\left(\frac{2}{1-\delta}\right)} \right)^{\frac{1}{4}} \quad (2)$$

*where* $\sigma^2 := \frac{1}{n} \sum_{i=1}^{n} \mathbb{E}\left[(y_i - g(\boldsymbol{s}_i, \boldsymbol{p}_i))^2\right]$ *and* $g(\boldsymbol{s}, \boldsymbol{p}) = \mathbb{E}\left[y \mid (\boldsymbol{s}, \boldsymbol{p})\right]$.

The proof of the above can be seen in Section 6. Note that the lowerbound proven here for $q$ is a necessary (and not a sufficient) condition for the required high probability (over data sampling) of the existence of a DeepONet with empirical risk below the threshold given above. For further insight, we now specialize our Theorem 4.1 to using $\mathcal{C} = 1$ – which then encompasses the case that we shall do experiments with, that of having DeepONets whose branch and trunk nets end in a sigmoid gate. Also, towards the following weakened bound – for a more intuitive presentation – we assume a common upperbound of $W$ on the 2−norm of the parameter vector for the branch and the trunk net and define $s := d_B + d_T$ as the upperbound on the total number of parameters in the predictor being trained.

**Theorem 4.2. (Lowerbounds for DeepONets Whose Branch and Trunk End in Sigmoid Gates)**
*Let* $\mathcal{C} = 1$ *and constants* $W$ *and* $s$ *be bounds on the norms of the weights of the branch and the trunk and the total number of trainable parameters respectively. Then* $\forall \delta \in (0,1)$, *and any arbitrary positive constant* $\epsilon > 0$ *if with probability at least* $1 - \delta$ *with respect to the sampling of the data* $\{(y_i, (\boldsymbol{s}_i, \boldsymbol{p}_i)) \mid i = 1, \ldots, n\}$, $\exists\, h_{\boldsymbol{w}_b, \boldsymbol{w}_t} \in \mathcal{H}$ *s.t,*

$$\frac{1}{n} \sum_{i=1}^{n} (y_i - h_{\boldsymbol{w}_b, \boldsymbol{w}_t}(\boldsymbol{s}_i, \boldsymbol{p}_i))^2 \leq \sigma^2 - \epsilon \left(1 + J \cdot (B + 2)\right)$$

*then,*

$$q \geq n^{\frac{1}{4}} \cdot \left( \frac{\epsilon^2}{288 \cdot B^2} \cdot \frac{1}{\ln\left(2 + e^{-s \cdot \alpha'} \cdot \left(\frac{4 \cdot \min\{d_B, d_T\}^2}{\epsilon} \cdot W \sqrt{s}\right)^s\right) + \ln\left(\frac{2}{1-\delta}\right)} \right)^{\frac{1}{4}} \quad (3)$$

*where* $\sigma^2 := \frac{1}{n} \sum_{i=1}^{n} \mathbb{E}\left[(y_i - g(\boldsymbol{s}_i, \boldsymbol{p}_i))^2\right]$ *and* $g(\boldsymbol{s}, \boldsymbol{p}) = \mathbb{E}\left[y \mid (\boldsymbol{s}, \boldsymbol{p})\right]$ *and if the branch net has* $\alpha$−*fraction of the training parameters then* $\alpha' = \frac{\alpha}{2} \ln \frac{1}{\alpha} + \frac{1-\alpha}{2} \ln \frac{1}{1-\alpha}$.

To interpret the above theorem consider a sequence of DeepONet training being done for fixed training data (and hence a fixed $n$) and on different architectures – but having the same weight bound and the same number of parameters and the same $q$, the common output dimension of the branch and the trunk functions. Now we can see how the above theorem reveals a largeness requirement for DeepONets - that if there has to

exist an architecture which can get the training error below a certain label-noise dependent threshold then necessarily the branch/trunk output dimension $q$ has to be $\Omega((\text{training--data--size})^{\frac{1}{4}})$.

Later, in Section 7, we shall conduct an experimental study motivated by the above and reveal something more than what the above theorem guarantees. We will see that over a sequence of training being done on different DeepONet architectures (and a fixed PDE) having nearly the same number of parameters, one can get monotonic improvement in performance upon increasing training data size $n$ if it is accompanied by an increase in $q$ s.t $\frac{q}{\sqrt{n}}$ is nearly constant. We also show that a slightly smaller rate of growth for $n$ for the same sequence of $q$s would break this monotonicity. Thus it reveals a "scaling law" for DeepONets - which is not yet within the ambit of our theoretical analysis.

## 5  Lemmas Towards Proving Theorem 4.1

**Lemma 5.1.** *For any space $X$ with Euclidean metric, we denote as $N(\theta, X)$ the covering number of it at scale $\theta$. Further recall from Definition 2, that $d_B$ and $d_T$ are the total number of parameters in any function in the sets $\mathcal{B}$ and $\mathcal{T}$ respectively. Let $\mathcal{W}_\mathcal{B} \subseteq \mathbb{R}^{d_B}$, $\mathcal{W}_\mathcal{T} \subseteq \mathbb{R}^{d_T}$ and $\mathcal{W}_\mathcal{H} = \mathcal{W}_\mathcal{B} \times \mathcal{W}_\mathcal{T}$ denote the sets of allowed weights of $\mathcal{B}$, $\mathcal{T}$, and $\mathcal{H}$ (Definition 3), respectively. Then the following three bounds hold for any $\theta > 0$,*

$$N(\theta, \mathcal{W}_\mathcal{B}) \le \left(\frac{2W_B\sqrt{d_B}}{\theta}\right)^{d_B} \qquad N(\theta, \mathcal{W}_\mathcal{T}) \le \left(\frac{2W_T\sqrt{d_T}}{\theta}\right)^{d_T}$$

$$N(\theta, \mathcal{W}_\mathcal{H}) \le N(\theta/2, \mathcal{W}_\mathcal{B}) \cdot N(\theta/2, \mathcal{W}_\mathcal{T})$$

The proof of the above Lemma is given in Section 5.1.1

**Lemma 5.2.** *We recall the definition of $\mathcal{H}$ from Definition 3, B as given in Defintion 1 & J from Definiton 4. Further for any $h$ and any training data of the form as given in Theorem 4.1, we denote the corresponding empirical risk as, $\hat{R}(h) := \frac{1}{n}\sum_{i=1}^{n}(y_i - h(\boldsymbol{s}_i, \boldsymbol{p}_i))^2$. Then, $\forall \theta > 0$ we have,*

$$\hat{\mathcal{R}}(h_{(\boldsymbol{w}_{b,\frac{\theta}{2}}, \boldsymbol{w}_{t,\frac{\theta}{2}})}) \le \hat{\mathcal{R}}(h_{(\boldsymbol{w}_b, \boldsymbol{w}_t)}) + q\mathcal{C}J\theta \cdot \left(B + 2q\mathcal{C}^2\right)$$

*and $\boldsymbol{w}_{b,\frac{\theta}{2}}$ and $\boldsymbol{w}_{t,\frac{\theta}{2}}$ be s.t. $\left\|\boldsymbol{w}_b - \boldsymbol{w}_{b,\frac{\theta}{2}}\right\| \le \frac{\theta}{2}$ and $\left\|\boldsymbol{w}_t - \boldsymbol{w}_{t,\frac{\theta}{2}}\right\| \le \frac{\theta}{2}$.*

Thus we see that it is quantifiable as to how much is the increment in the empirical risk when for a given training data a DeepONet is replaced by another with weights within a distance of $\theta$ from the original - and that this increment is parametric in $\theta$. The proof of the above lemma is given in Section 5.1.2.

**Lemma 5.3.** *We recall the definition of $\mathcal{H}_\theta$ from Definition 3; $d_B$, $d_T$, $W_B$, $W_T$, $\mathcal{C}$ & $q$ from Defintion 2 and B as given in Defintion 1. Then $\forall \theta > 0$, and for $z_i := y_i - g(\boldsymbol{s}_i, \boldsymbol{p}_i)$;*

$$\mathbb{P}\left(\exists\, h_{(\boldsymbol{w}_{b,\frac{\theta}{2}}, \boldsymbol{w}_{t,\frac{\theta}{2}})} \in \mathcal{H}_\theta \mid \frac{1}{n}\sum_{i=1}^{n} h_{(\boldsymbol{w}_{b,\frac{\theta}{2}}, \boldsymbol{w}_{t,\frac{\theta}{2}})}(\boldsymbol{s}_i, \boldsymbol{p}_i)\, z_i \ge \frac{\theta}{4}\right)$$

$$\le \frac{2^{2(d_B+d_T)+1}}{\theta^{(d_B+d_T)}} \cdot \left(W_B\sqrt{d_B}\right)^{d_B} \cdot \left(W_T\sqrt{d_T}\right)^{d_T} \cdot \exp\left(-\frac{2n\theta^2}{8^4 \cdot (B \cdot q\mathcal{C}^2)^2}\right)$$

$$+ 2\exp\left(\frac{-n\theta^2}{8^3 \cdot B^2 \cdot q^2 \cdot \mathcal{C}^4}\right)$$

The proof of the above lemma is given in Section 5.1.3

**Lemma 5.4.** *We continue in the same setup as in the previous lemma and further recall the definition of $\sigma$ as in Theorem 4.1. Then $\forall \theta > 0$*

$$\mathbb{P}\left(\exists\ h_{\boldsymbol{w}_b,\boldsymbol{w}_t} \in \mathcal{H} \mid \frac{1}{n}\sum_{i=1}^n \left(y_i - h_{\boldsymbol{w}_b,\boldsymbol{w}_t}(\boldsymbol{s}_i,\boldsymbol{p}_i)\right)^2 \le \sigma^2 - \theta\right) \le 2\exp\left(-\frac{n\theta^2}{288B^2}\right) + \mathbb{P}\left(\exists\ h_{\boldsymbol{w}_b,\boldsymbol{w}_t} \in \mathcal{H} \mid \frac{1}{n}\sum_{i=1}^n h\left(\boldsymbol{s}_i,\boldsymbol{p}_i\right)z_i \ge \frac{\theta}{4}\right)$$

The above lemma reveals an intimate connection between the empirical error of DeepONets and the correlation of its output with label noise. The proof of the above lemma is given in Section 5.1.4

## 5.1 Proofs of the Lemmas

In the following sections we give the proofs of the above listed lemmas.

### 5.1.1 Proof of Lemma 5.1

*Proof.* The first two equations are standard results, Example 27.1 of (Shalev-Shwartz & Ben-David, 2014)

Further define $d(\boldsymbol{x},\boldsymbol{y}) = \|\boldsymbol{x}-\boldsymbol{y}\|_2$. Then, let $S \subset \mathbb{R}^{d_B}$ be a witness for $N(\theta/2,\mathcal{W}_{\mathcal{B}})$, that is, for all $\boldsymbol{w}_b \in \mathcal{W}_{\mathcal{B}}$, there is some $s \in S$ such that $d(\boldsymbol{w}_b,s) \le \theta/2$. Similarly, let $P \subset \mathbb{R}^{d_T}$ be a witness for $N(\theta/2,\mathcal{W}_{\mathcal{T}})$. Then for all $\boldsymbol{w}_b \in \mathcal{W}_{\mathcal{B}}$, $\boldsymbol{w}_t \in \mathcal{W}_{\mathcal{T}}$, there exist a corresponding cover point $s \in S$ and $p \in P$. And since $(\boldsymbol{w}_b,\boldsymbol{w}_t) \in \mathcal{W}_{\mathcal{H}}$:

$$\begin{aligned}
d((\boldsymbol{w}_b,\boldsymbol{w}_t),(s,p)) &\le d((\boldsymbol{w}_b,\boldsymbol{w}_t),(s,\boldsymbol{w}_t)) + d((s,\boldsymbol{w}_t),(s,p)) \quad \text{(by triangle inequality)} \\
&= d(\boldsymbol{w}_b,s) + d(\boldsymbol{w}_t,p) \quad \text{(under } d \sim l_2\text{-norm)} \\
&\le \theta \quad \text{(by definition of } S \text{ and } P)
\end{aligned}$$

Hence, $S \times T$ is an $\theta$-cover of $\mathcal{W}_{\mathcal{H}}$.

$\square$

### 5.1.2 Proof of Lemma 5.2

*Proof.* Given an $\theta > 0$ and a $h_{(\boldsymbol{w}_b,\boldsymbol{w}_t)} \in \mathcal{H}$, let $\boldsymbol{w}_{b,\frac{\theta}{2}}$ and $\boldsymbol{w}_{t,\frac{\theta}{2}}$ be s.t. $\left\|\boldsymbol{w}_b - \boldsymbol{w}_{b,\frac{\theta}{2}}\right\| \le \frac{\theta}{2}$ and $\left\|\boldsymbol{w}_t - \boldsymbol{w}_{t,\frac{\theta}{2}}\right\| \le \frac{\theta}{2}$. Then from the definition of $J$ in Definition 4, the following inequalities hold,

$$\sup_{\boldsymbol{s}} \left\|B_{\boldsymbol{w}_b}(\boldsymbol{s}) - B_{\boldsymbol{w}_{b,\frac{\theta}{2}}}(\boldsymbol{s})\right\|_\infty \le J.\frac{\theta}{2} \text{ and } \sup_{\boldsymbol{p}} \left\|T_{\boldsymbol{w}_t}(\boldsymbol{p}) - T_{\boldsymbol{w}_{t,\frac{\theta}{2}}}(\boldsymbol{p})\right\|_\infty \le J.\frac{\theta}{2}$$

Further, we can simplify as follows, for any valid $(\boldsymbol{s},\boldsymbol{p})$ input to the function $h_{\boldsymbol{w}_b,\boldsymbol{w}_t} = \langle B_{\boldsymbol{w}_b}, T_{\boldsymbol{w}_t}\rangle$ and similarly for $h_{\boldsymbol{w}_{b,\frac{\theta}{2}},\boldsymbol{w}_{t,\frac{\theta}{2}}}$.

$$\begin{aligned}
&\left|\langle B_{\boldsymbol{w}_b}(\boldsymbol{s}), T_{\boldsymbol{w}_t}(\boldsymbol{p})\rangle - \left\langle B_{\boldsymbol{w}_{b,\frac{\theta}{2}}}(\boldsymbol{s}), T_{\boldsymbol{w}_{t,\frac{\theta}{2}}}(\boldsymbol{p})\right\rangle\right| \\
=&\left|\langle B_{\boldsymbol{w}_b}(\boldsymbol{s}), T_{\boldsymbol{w}_t}(\boldsymbol{p})\rangle - \left\langle B_{\boldsymbol{w}_b}(\boldsymbol{s}), T_{\boldsymbol{w}_{t,\frac{\theta}{2}}}(\boldsymbol{p})\right\rangle + \left\langle B_{\boldsymbol{w}_b}(\boldsymbol{s}), T_{\boldsymbol{w}_{t,\frac{\theta}{2}}}(\boldsymbol{p})\right\rangle - \left\langle B_{\boldsymbol{w}_{b,\frac{\theta}{2}}}(\boldsymbol{s}), T_{\boldsymbol{w}_{t,\frac{\theta}{2}}}(\boldsymbol{p})\right\rangle\right| \\
\le&\left|\left\langle B_{\boldsymbol{w}_b}(\boldsymbol{s}), T_{\boldsymbol{w}_t}(\boldsymbol{p}) - T_{\boldsymbol{w}_{t,\frac{\theta}{2}}}(\boldsymbol{p})\right\rangle\right| + \left|\left\langle T_{\boldsymbol{w}_{t,\frac{\theta}{2}}}(\boldsymbol{p}), B_{\boldsymbol{w}_b}(\boldsymbol{s}) - B_{\boldsymbol{w}_{b,\frac{\theta}{2}}}(\boldsymbol{s})\right\rangle\right|
\end{aligned}$$

To upperbound the above we recall (a) the definition of $\mathcal{C}$ from Definitions 2 and (b) that for any two $q$–dimensional vectors $\boldsymbol{a}$ and $\boldsymbol{b}$ we have, $|\langle \boldsymbol{a},\boldsymbol{b}\rangle| \le \sum_{i=1}^q |a_i||b_i| \le (\max_{i=1,\ldots,q}|b_i|)\sum_{i=1}^q |a_i|$. Thus we have,

$$\forall (\boldsymbol{s},\boldsymbol{p}),\ \left|\langle B_{\boldsymbol{w}_b}(\boldsymbol{s}), T_{\boldsymbol{w}_t}(\boldsymbol{p})\rangle - \left\langle B_{\boldsymbol{w}_{b,\frac{\theta}{2}}}(\boldsymbol{s}), T_{\boldsymbol{w}_{t,\frac{\theta}{2}}}(\boldsymbol{p})\right\rangle\right| \le 2\cdot\left(\frac{J\theta}{2}\cdot q\cdot\mathcal{C}\right) \tag{4}$$

$$\implies \forall (\boldsymbol{s},\boldsymbol{p}),\ \left|h_{\boldsymbol{w}_b,\boldsymbol{w}_t}(\boldsymbol{s},\boldsymbol{p}) - h_{\boldsymbol{w}_{b,\frac{\theta}{2}},\boldsymbol{w}_{t,\frac{\theta}{2}}}(\boldsymbol{s},\boldsymbol{p})\right| \le q\cdot\mathcal{C}\cdot J\theta \tag{5}$$

Define, $r_{1,i} := \left(y_i - h_{(\boldsymbol{w}_{b,\frac{\theta}{2}}, \boldsymbol{w}_{t,\frac{\theta}{2}})}(\boldsymbol{s}_i, \boldsymbol{p}_i)\right)$ and $r_{2,i} := \left(y_i - h_{(\boldsymbol{w}_b, \boldsymbol{w}_t)}(\boldsymbol{s}_i, \boldsymbol{p}_i)\right)$

Now,

$$
\begin{aligned}
r_{1,i}^2 - r_{2,i}^2 &= (h_{(\boldsymbol{w}_{b,\frac{\theta}{2}}, \boldsymbol{w}_{t,\frac{\theta}{2}})}(\boldsymbol{s}_i, \boldsymbol{p}_i)^2 - h_{(\boldsymbol{w}_b, \boldsymbol{w}_t)}(\boldsymbol{s}_i, \boldsymbol{p}_i)^2) + 2y_i\left(h_{(\boldsymbol{w}_b, \boldsymbol{w}_t)}(\boldsymbol{s}_i, \boldsymbol{p}_i) - h_{(\boldsymbol{w}_{b,\frac{\theta}{2}}, \boldsymbol{w}_{t,\frac{\theta}{2}})}(\boldsymbol{s}_i, \boldsymbol{p}_i)\right) \\
&\le \left(\left|h_{(\boldsymbol{w}_b, \boldsymbol{w}_t)}(\boldsymbol{s}_i, \boldsymbol{p}_i) - h_{(\boldsymbol{w}_{b,\frac{\theta}{2}}, \boldsymbol{w}_{t,\frac{\theta}{2}})}(\boldsymbol{s}_i, \boldsymbol{p}_i)\right|\right)\left(h_{(\boldsymbol{w}_b, \boldsymbol{w}_t)}(\boldsymbol{s}_i, \boldsymbol{p}_i) + h_{(\boldsymbol{w}_{b,\frac{\theta}{2}}, \boldsymbol{w}_{t,\frac{\theta}{2}})}(\boldsymbol{s}_i, \boldsymbol{p}_i)\right) + 2 \cdot B \cdot q \cdot \mathcal{C} \cdot J\theta \\
&\le \left(h_{(\boldsymbol{w}_b, \boldsymbol{w}_t)}(\boldsymbol{s}_i, \boldsymbol{p}_i) + h_{(\boldsymbol{w}_{b,\frac{\theta}{2}}, \boldsymbol{w}_{t,\frac{\theta}{2}})}(\boldsymbol{s}_i, \boldsymbol{p}_i)\right) \cdot q \cdot \mathcal{C} \cdot J\theta + B \cdot q \cdot \mathcal{C} \cdot J\theta \\
&\le q \cdot \mathcal{C} \cdot J\theta \cdot \left((h_{(\boldsymbol{w}_b, \boldsymbol{w}_t)}(\boldsymbol{s}_i, \boldsymbol{p}_i) + h_{(\boldsymbol{w}_{b,\frac{\theta}{2}}, \boldsymbol{w}_{t,\frac{\theta}{2}})}(\boldsymbol{s}_i, \boldsymbol{p}_i) + B\right)
\end{aligned}
$$

Averaging the above over all training data we get,

$$
\frac{1}{n}\sum_{i=1}^n r_{1,i}^2 \le \frac{1}{n}\sum_{i=1}^n r_{2,i}^2 + \frac{1}{n}\sum_{i=1}^n q \cdot \mathcal{C} \cdot J\theta \cdot \left((h_{(\boldsymbol{w}_b, \boldsymbol{w}_t)}(\boldsymbol{s}_i, \boldsymbol{p}_i) + h_{(\boldsymbol{w}_{b,\frac{\theta}{2}}, \boldsymbol{w}_{t,\frac{\theta}{2}})}(\boldsymbol{s}_i, \boldsymbol{p}_i)) + B\right) \tag{6}
$$

Using Cauchy-Schwarz over the inner-product in the definition of $h$, we get,

$$
\left|h_{(w_b, w_t)}(s_i, p_i)\right| \le \sqrt{q}\mathcal{C} \cdot \sqrt{q}\mathcal{C} \le q \cdot \mathcal{C}^2 \implies \left((h_{(\boldsymbol{w}_b, \boldsymbol{w}_t)}(\boldsymbol{s}_i, \boldsymbol{p}_i) + h_{(\boldsymbol{w}_{b,\frac{\theta}{2}}, \boldsymbol{w}_{t,\frac{\theta}{2}})}(\boldsymbol{s}_i, \boldsymbol{p}_i))\right) \le 2q \cdot \mathcal{C}^2 \tag{7}
$$

Substituting the above into equation 6 and invoking the definition of $\hat{\mathcal{R}}$,

$$
\begin{aligned}
\hat{\mathcal{R}}(h_{(\boldsymbol{w}_{b,\frac{\theta}{2}}, \boldsymbol{w}_{t,\frac{\theta}{2}})}) &\le \hat{\mathcal{R}}(h_{(\boldsymbol{w}_b, \boldsymbol{w}_t)}) + \frac{1}{n}\sum_{i=1}^n q \cdot \mathcal{C} \cdot J\theta \cdot \left((h_{(\boldsymbol{w}_b, \boldsymbol{w}_t)}(\boldsymbol{s}_i, \boldsymbol{p}_i) + h_{(\boldsymbol{w}_{b,\frac{\theta}{2}}, \boldsymbol{w}_{t,\frac{\theta}{2}})}(\boldsymbol{s}_i, \boldsymbol{p}_i)) + B\right) \\
&\le \hat{\mathcal{R}}(h_{(\boldsymbol{w}_b, \boldsymbol{w}_t)}) + (q \cdot \mathcal{C} \cdot J\theta \cdot B) + (q \cdot \mathcal{C} \cdot J\theta) \cdot \left(2q \cdot \mathcal{C}^2\right) \\
&\le \hat{\mathcal{R}}(h_{(\boldsymbol{w}_b, \boldsymbol{w}_t)}) + q\mathcal{C}J\theta \cdot \left(B + 2q\mathcal{C}^2\right)
\end{aligned}
$$

The above is what we set out to prove.

$\square$

### 5.1.3   Proof of Lemma 5.3

*Proof.* Recall that for each data $i$, we had defined the random variable, $z_i := y_i - g(\boldsymbol{s}_i, \boldsymbol{p}_i)$. Since $g(\boldsymbol{s}, \boldsymbol{p}) = \mathbb{E}[y \mid (\boldsymbol{s}, \boldsymbol{p})]$, we can note that $\mathbb{E}[z_i] = 0$. Further,

$$
z_i^2 = (y_i - g(\boldsymbol{s}_i, \boldsymbol{p}_i))^2 \le y_i^2 - 2 \cdot y_i \cdot g(\boldsymbol{s}_i, \boldsymbol{p}_i) + g(\boldsymbol{s}_i, \boldsymbol{p}_i)^2 \le 4B^2 \tag{8}
$$

Recall from equation 7. that $\left|h_{(\boldsymbol{w}_{b,\frac{\theta}{2}}, \boldsymbol{w}_{t,\frac{\theta}{2}})}(\boldsymbol{s}_i, \boldsymbol{p}_i)\right| \le q \cdot \mathcal{C}^2$

For each data $i$, we further define the random variable, $Y_{\theta,i} := \left((h_{(\boldsymbol{w}_{b,\frac{\theta}{2}}, \boldsymbol{w}_{t,\frac{\theta}{2}})}(\boldsymbol{s}_i, \boldsymbol{p}_i) - \mathbb{E}[h_{(\boldsymbol{w}_{b,\frac{\theta}{2}}, \boldsymbol{w}_{t,\frac{\theta}{2}})}])z_i\right)$

Now note that,

$$\mathbb{E}[Y_{\theta,i}] = \mathbb{E}\left[\left(h_{(\boldsymbol{w}_{b,\frac{\theta}{2}},\boldsymbol{w}_{t,\frac{\theta}{2}})}(\boldsymbol{s}_i,\boldsymbol{p}_i) - \mathbb{E}[h_{(\boldsymbol{w}_{b,\frac{\theta}{2}},\boldsymbol{w}_{t,\frac{\theta}{2}})}]\right)z_i\right]$$

$$= \mathbb{E}\left[h_{(\boldsymbol{w}_{b,\frac{\theta}{2}},\boldsymbol{w}_{t,\frac{\theta}{2}})}(\boldsymbol{s}_i,\boldsymbol{p}_i)\cdot y_i\right] - \mathbb{E}\left[h_{(\boldsymbol{w}_{b,\frac{\theta}{2}},\boldsymbol{w}_{t,\frac{\theta}{2}})}(\boldsymbol{s}_i,\boldsymbol{p}_i)\cdot g(\boldsymbol{s}_i,\boldsymbol{p}_i)\right]$$

Next, we use the tower property of conditional expectation to expand the first term,

$$\mathbb{E}\left[h_{(\boldsymbol{w}_{b,\frac{\theta}{2}},\boldsymbol{w}_{t,\frac{\theta}{2}})}(\boldsymbol{s}_i,\boldsymbol{p}_i)\cdot y_i\right] = \mathbb{E}\left[\mathbb{E}[h_{(\boldsymbol{w}_{b,\frac{\theta}{2}},\boldsymbol{w}_{t,\frac{\theta}{2}})}(\boldsymbol{s}_i,\boldsymbol{p}_i)\,y_i\mid(\boldsymbol{s}_i,\boldsymbol{p}_i)]\right] = \mathbb{E}[h_{(\boldsymbol{w}_{b,\frac{\theta}{2}},\boldsymbol{w}_{t,\frac{\theta}{2}})}(\boldsymbol{s}_i,\boldsymbol{p}_i)\,\mathbb{E}[y\mid(\boldsymbol{s}_i,\boldsymbol{p}_i)]]$$

$$= \mathbb{E}\left[h_{(\boldsymbol{w}_{b,\frac{\theta}{2}},\boldsymbol{w}_{t,\frac{\theta}{2}})}(\boldsymbol{s}_i,\boldsymbol{p}_i)\cdot g(\boldsymbol{s}_i,\boldsymbol{p}_i)\right]$$

Substituting this back into the previous equation, we get,

$$\mathbb{E}[Y_{\theta,i}] = 0$$

Further,

$$|Y_{\theta,i}| = \left|h_{(\boldsymbol{w}_{b,\frac{\theta}{2}},\boldsymbol{w}_{t,\frac{\theta}{2}})}(\boldsymbol{s}_i,\boldsymbol{p}_i) - \mathbb{E}[h_{(\boldsymbol{w}_{b,\frac{\theta}{2}},\boldsymbol{w}_{t,\frac{\theta}{2}})}]\right|\cdot|z_i| \leq \left(\left|h_{(\boldsymbol{w}_{b,\frac{\theta}{2}},\boldsymbol{w}_{t,\frac{\theta}{2}})}(\boldsymbol{s}_i,\boldsymbol{p}_i)\right| + \left|\mathbb{E}[h_{(\boldsymbol{w}_{b,\frac{\theta}{2}},\boldsymbol{w}_{t,\frac{\theta}{2}})}]\right|\right)\cdot 2B$$

$$\leq 4\cdot\mathcal{C}^2\cdot B\cdot q$$

Applying Hoeffding's inequality [1] on $Y_{\theta,i}$, we will get,

$$\mathbb{P}\left(\frac{1}{n}\sum_{i=1}^n\left((h_{(\boldsymbol{w}_{b,\frac{\theta}{2}},\boldsymbol{w}_{t,\frac{\theta}{2}})}(\boldsymbol{s}_i,\boldsymbol{p}_i) - \mathbb{E}[h_{(\boldsymbol{w}_{b,\frac{\theta}{2}},\boldsymbol{w}_{t,\frac{\theta}{2}})}])z_i\right)\geq t\right) \leq \exp\left(-\frac{2nt^2}{(8\cdot B\cdot q\mathcal{C}^2)^2}\right) \tag{9}$$

We choose $t = \frac{\theta}{8}$ to get,

$$\mathbb{P}\left(\left|\frac{1}{n}\sum_{i=1}^n\left((h_{(\boldsymbol{w}_{b,\frac{\theta}{2}},\boldsymbol{w}_{t,\frac{\theta}{2}})}(\boldsymbol{s}_i,\boldsymbol{p}_i) - \mathbb{E}[h_{(\boldsymbol{w}_{b,\frac{\theta}{2}},\boldsymbol{w}_{t,\frac{\theta}{2}})}])z_i\right)\right| \geq \frac{\theta}{8}\right) \leq 2\cdot\exp\left(-\frac{2n\theta^2}{8^4\cdot(B\cdot q\mathcal{C}^2)^2}\right)$$

We define two events,

---

[1]

**Theorem 5.5.** *(Hoeffding's inequality).* *Let* $Z_1,\ldots,Z_n$ *be independent bounded random variables with* $Z_i \in [a,b]$ *for all* $i$, *where* $-\infty < a \leq b < \infty$. *Then*

$$\mathbb{P}\left(\frac{1}{n}\sum_{i=1}^n(Z_i - \mathbb{E}[Z_i]) \geq t\right) \leq \exp\left(-\frac{2nt^2}{(b-a)^2}\right)$$

*and*

$$\mathbb{P}\left(\frac{1}{n}\sum_{i=1}^n(Z_i - \mathbb{E}[Z_i]) \leq -t\right) \leq \exp\left(-\frac{2nt^2}{(b-a)^2}\right)$$

*for all* $t \geq 0$.

$$\mathbf{E}_5 \coloneqq \left\{ \left| \frac{1}{n} \sum_{i=1}^{n} z_i \right| \geq \frac{\theta}{8 \cdot q\mathcal{C}^2} \right\} \;\&\; \mathbf{E}_6 \coloneqq \left\{ \exists\, h_{(\boldsymbol{w}_{b,\frac{\theta}{2}}, \boldsymbol{w}_{t,\frac{\theta}{2}})} \in \mathcal{H}_\theta \mid \frac{1}{n} \sum_{i=1}^{n} \mathbb{E}[h_{(\boldsymbol{w}_{b,\frac{\theta}{2}}, \boldsymbol{w}_{t,\frac{\theta}{2}})}] z_i \geq \frac{\theta}{8} \right\}$$

Recalling the bound on the $h$ function we have, $\frac{1}{q \cdot \mathcal{C}^2} \cdot |\mathbb{E}[h_{(\boldsymbol{w}_{b,\frac{\theta}{2}}, \boldsymbol{w}_{t,\frac{\theta}{2}})}]| \in [0,1]$, we have that if $\mathbf{E}_5^c$ happens then for such a sample of $\{z_i, i = 1, \ldots, n\}$,

$$\forall h_{(\boldsymbol{w}_{b,\frac{\theta}{2}}, \boldsymbol{w}_{t,\frac{\theta}{2}})} \in \mathcal{H}_\theta,$$
$$\frac{\theta}{8 \cdot q\mathcal{C}^2} > \left| \frac{1}{n} \sum_{i=1}^{n} z_i \right| \geq \frac{1}{q \cdot \mathcal{C}^2} \cdot |\mathbb{E}[h_{(\boldsymbol{w}_{b,\frac{\theta}{2}}, \boldsymbol{w}_{t,\frac{\theta}{2}})}]| \left| \frac{1}{n} \sum_{i=1}^{n} z_i \right| \geq \frac{1}{n \cdot q\mathcal{C}^2} \sum_{i=1}^{n} \mathbb{E}[h_{(\boldsymbol{w}_{b,\frac{\theta}{2}}, \boldsymbol{w}_{t,\frac{\theta}{2}})}] z_i$$

Hence $\mathbf{E}_5^c \implies \mathbf{E}_6^c$ and hence $\mathbb{P}(\mathbf{E}_6) \leq \mathbb{P}(\mathbf{E}_5)$ i.e

$$\mathbb{P}\left( \exists\, h_{(\boldsymbol{w}_{b,\frac{\theta}{2}}, \boldsymbol{w}_{t,\frac{\theta}{2}})} \in \mathcal{H} \mid \frac{1}{n} \sum_{i=1}^{n} \mathbb{E}[h_{(\boldsymbol{w}_{b,\frac{\theta}{2}}, \boldsymbol{w}_{t,\frac{\theta}{2}})}] z_i \geq \frac{\theta}{8} \right) \leq \mathbb{P}\left( \left| \frac{1}{n} \sum_{i=1}^{n} z_i \right| \geq \frac{\theta}{8 \cdot q \cdot \mathcal{C}^2} \right) \tag{10}$$

Recalling that $(|z_i| \leq 2B)$, by Hoeffding's inequality we have,

$$\mathbb{P}\left( \left| \frac{1}{n} \sum_{i=1}^{n} z_i \right| \geq \frac{\theta}{8 \cdot q \cdot \mathcal{C}^2} \right) \leq 2\exp\left( \frac{-n\theta^2}{8^3 \cdot B^2 \cdot q^2 \cdot \mathcal{C}^4} \right) \tag{11}$$

Now we define three events $\mathbf{E}_7, \mathbf{E}_8$ and $\mathbf{E}_9$ as follows,

$$\mathbf{E}_7 \coloneqq \left\{ \forall\, h_{(\boldsymbol{w}_{b,\frac{\theta}{2}}, \boldsymbol{w}_{t,\frac{\theta}{2}})} \in \mathcal{H}_\theta, \frac{1}{n} \sum_{i=1}^{n} \left( h_{(\boldsymbol{w}_{b,\frac{\theta}{2}}, \boldsymbol{w}_{t,\frac{\theta}{2}})}(\boldsymbol{s}_i, \boldsymbol{p}_i) - \mathbb{E}[h_{(\boldsymbol{w}_{b,\frac{\theta}{2}}, \boldsymbol{w}_{t,\frac{\theta}{2}})}] \right) z_i \leq \frac{\theta}{8} \right\}$$

$$\mathbf{E}_8 \coloneqq \left\{ \forall\, h_{(\boldsymbol{w}_{b,\frac{\theta}{2}}, \boldsymbol{w}_{t,\frac{\theta}{2}})} \in \mathcal{H}_\theta, \frac{1}{n} \sum_{i=1}^{n} \mathbb{E}[h_{(\boldsymbol{w}_{b,\frac{\theta}{2}}, \boldsymbol{w}_{t,\frac{\theta}{2}})}] z_i \leq \frac{\theta}{8} \right\}$$

$$\mathbf{E}_9 \coloneqq \left\{ \forall\, h_{(\boldsymbol{w}_{b,\frac{\theta}{2}}, \boldsymbol{w}_{t,\frac{\theta}{2}})} \in \mathcal{H}_\theta, \frac{1}{n} \sum_{i=1}^{n} h_{(\boldsymbol{w}_{b,\frac{\theta}{2}}, \boldsymbol{w}_{t,\frac{\theta}{2}})}(\boldsymbol{s}_i, \boldsymbol{p}_i) z_i \leq \frac{\theta}{4} \right\}$$

Observe that, if $\mathbf{E}_7$ and $\mathbf{E}_8$ hold then $\mathbf{E}_9$ will also hold.

Hence,

$$\mathbb{P}(\mathbf{E}_7 \cap \mathbf{E}_8) \leq \mathbb{P}(\mathbf{E}_9) \implies \mathbb{P}(\mathbf{E}_9^c) \leq \mathbb{P}(\mathbf{E}_7^c) + \mathbb{P}(\mathbf{E}_8^c)$$

Thus, we can invoke equations 10 and 11 to get,

$$\mathbb{P}\left( \exists h_{(\boldsymbol{w}_{b,\frac{\theta}{2}}, \boldsymbol{w}_{t,\frac{\theta}{2}})} \in \mathcal{H}_\theta \mid \frac{1}{n} \sum_{i=1}^{n} h_{(\boldsymbol{w}_{b,\frac{\theta}{2}}, \boldsymbol{w}_{t,\frac{\theta}{2}})}(\boldsymbol{s}_i, \boldsymbol{p}_i) z_i \geq \frac{\theta}{4} \right)$$

$$
\leq \mathbb{P}\left(\exists h_{(\boldsymbol{w}_{b,\frac{\theta}{2}},\boldsymbol{w}_{t,\frac{\theta}{2}})} \in \mathcal{H}_\theta \mid \frac{1}{n}\left|\sum_{i=1}^{n}\left(h_{(\boldsymbol{w}_{b,\frac{\theta}{2}},\boldsymbol{w}_{t,\frac{\theta}{2}})}(\boldsymbol{s}_i,\boldsymbol{p}_i) - \mathbb{E}[h_{(\boldsymbol{w}_{b,\frac{\theta}{2}},\boldsymbol{w}_{t,\frac{\theta}{2}})}]\right)z_i\right| \geq \frac{\theta}{8}\right) + \mathbb{P}\left(\left|\frac{1}{n}\sum_{i=1}^{n}z_i\right| \geq \frac{\theta}{8 \cdot q \cdot \mathcal{C}^2}\right)
$$

$$
\leq \mathbb{P}\left(\bigcup_{h_{(\boldsymbol{w}_{b,\frac{\theta}{2}},\boldsymbol{w}_{t,\frac{\theta}{2}})}\in\mathcal{H}_\theta}\left\{\frac{1}{n}\left|\sum_{i=1}^{n}(h_{(\boldsymbol{w}_{b,\frac{\theta}{2}},\boldsymbol{w}_{t,\frac{\theta}{2}})}(\boldsymbol{s}_i,\boldsymbol{p}_i) - \mathbb{E}[h_{(\boldsymbol{w}_{b,\frac{\theta}{2}},\boldsymbol{w}_{t,\frac{\theta}{2}})}])z_i\right| \geq \frac{\theta}{8}\right\}\right) + 2\exp\left(\frac{-n\theta^2}{8^3 \cdot B^2 \cdot q^2 \cdot \mathcal{C}^4}\right)
$$

$$
\leq \sum_{h_{(\boldsymbol{w}_{b,\frac{\theta}{2}},\boldsymbol{w}_{t,\frac{\theta}{2}})}\in\mathcal{H}_\theta}\mathbb{P}\left(\frac{1}{n}\left|\sum_{i=1}^{n}(h_{(\boldsymbol{w}_{b,\frac{\theta}{2}},\boldsymbol{w}_{t,\frac{\theta}{2}})}(\boldsymbol{s}_i,\boldsymbol{p}_i) - \mathbb{E}[h_{(\boldsymbol{w}_{b,\frac{\theta}{2}},\boldsymbol{w}_{t,\frac{\theta}{2}})}])z_i\right| \geq \frac{\theta}{8}\right) + 2\exp\left(\frac{-n\theta^2}{8^3 \cdot B^2 \cdot q^2 \cdot \mathcal{C}^4}\right)
$$

Hence,

$$
\mathbb{P}\left(\exists h_{(\boldsymbol{w}_{b,\frac{\theta}{2}},\boldsymbol{w}_{t,\frac{\theta}{2}})} \in \mathcal{H}_\theta \mid \frac{1}{n}\sum_{i=1}^{n}h_{(\boldsymbol{w}_{b,\frac{\theta}{2}},\boldsymbol{w}_{t,\frac{\theta}{2}})}(\boldsymbol{s}_i,\boldsymbol{p}_i)z_i \geq \frac{\theta}{4}\right)
$$

$$
\leq \frac{2^{2(d_B+d_T)}}{\theta^{(d_B+d_T)}} \cdot \left(W_B\sqrt{d_B}\right)^{d_B} \cdot \left(W_T\sqrt{d_T}\right)^{d_T} \cdot 2 \cdot \exp\left(-\frac{2n\theta^2}{8^4 \cdot (B \cdot q\mathcal{C}^2)^2}\right)
$$

$$
+ 2\exp\left(\frac{-n\theta^2}{8^3 \cdot B^2 \cdot q^2 \cdot \mathcal{C}^4}\right)
$$

And the above is what we set out to prove. □

### 5.1.4 Proof of Lemma 5.4

*Proof.* Recall the definition of $z_i$ from the previous proof and and from the assumptions in Theorem 4.2 we have, $\frac{1}{n}\sum_{i=1}^{n}\mathbb{E}\left[z_i^2\right] = \sigma^2$. Recalling that $z_i \in [-2B, 2B]$ and they are i.i.d. we can invoke Hoeffding's Lemma 5.5 (with $t = \frac{\theta}{6}, b = 2B, a = -2B$) to get,

$$
\mathbb{P}\left(\frac{1}{n}\sum_{i=1}^{n}z_i^2 \leq \sigma^2 - \frac{\theta}{6}\right) \leq \exp\left(-\frac{n\theta^2}{288B^2}\right) \tag{12}
$$

Further note that, $z_i \cdot g(\boldsymbol{s}_i, \boldsymbol{p}_i)$ is i.i.d with mean 0 since $\mathbb{E}[z_i \mid (\boldsymbol{s}_i, \boldsymbol{p}_i)] = 0$ and $|z_i \cdot g(\boldsymbol{s}_i, \boldsymbol{p}_i)| \leq 2B$ Applying Hoeffding's inequality again,

$$
\mathbb{P}\left(\frac{1}{n}\sum_{i=1}^{n}z_i g(\boldsymbol{s}_i, \boldsymbol{p}_i) \leq -\frac{\theta}{6}\right) \leq \exp\left(-\frac{n\theta^2}{288B^2}\right) \tag{13}
$$

Given a $h_{\boldsymbol{w}_b,\boldsymbol{w}_t} \in \mathcal{H}$, we define the following vector random variables,

$$
Z := \frac{1}{\sqrt{n}}(z_1, z_2, \cdots, z_n) \tag{14}
$$

$$
G = \frac{1}{\sqrt{n}}(g(\boldsymbol{s}_1, \boldsymbol{p}_1), g(\boldsymbol{s}_2, \boldsymbol{p}_2), \cdots, g(\boldsymbol{s}_n, \boldsymbol{p}_n)) \tag{15}
$$

$$
F = \frac{1}{\sqrt{n}}(h_{\boldsymbol{w}_b,\boldsymbol{w}_t}(\boldsymbol{s}_1, \boldsymbol{p}_1), h_{\boldsymbol{w}_b,\boldsymbol{w}_t}(\boldsymbol{s}_2, \boldsymbol{p}_2), \cdots, h_{\boldsymbol{w}_b,\boldsymbol{w}_t}(\boldsymbol{s}_n, \boldsymbol{p}_n)) \tag{16}
$$

Note that,

$$\|G + Z - F\|^2 = \|\frac{1}{\sqrt{n}}\left(g\left(\boldsymbol{s}_1, \boldsymbol{p}_1\right), \cdots, g\left(\boldsymbol{s}_n, \boldsymbol{p}_n\right)\right) + \frac{1}{\sqrt{n}}\left(z_1, \ldots, z_n\right) - \frac{1}{\sqrt{n}}(h_{\boldsymbol{w}_b, \boldsymbol{w}_t}\left(\boldsymbol{s}_1, \boldsymbol{p}_1\right), \cdots h_{\boldsymbol{w}_b, \boldsymbol{w}_t}\left(\boldsymbol{s}_n, \boldsymbol{p}_n\right))\|^2 \tag{17}$$

Recalling that $z_i := y_i - g\left(\boldsymbol{s}_i, \boldsymbol{p}_i\right)$ and the definition of the empirical risk of the predictor, $\hat{R}(h_{\boldsymbol{w}_b, \boldsymbol{w}_t}) := \frac{1}{n}\sum_{i=1}^{n}(y_i - h_{\boldsymbol{w}_b, \boldsymbol{w}_t}(\boldsymbol{s}_i, \boldsymbol{p}_i))^2$, we realize that,

$$\|Z + G - F\|^2 = \hat{R}(h_{\boldsymbol{w}_b, \boldsymbol{w}_t})$$

Suppose, $\|Z\|^2 \geqslant \sigma^2 - \frac{\theta}{6}$ and $\langle Z, G \rangle \geqslant -\frac{\theta}{6}$. Then we have,

$$\|Z + G - F\|^2 = \|Z\|^2 + 2\langle Z, G - F \rangle + \|G - F\|^2 = \|Z\|^2 + 2\langle Z, G \rangle - 2\langle Z, F \rangle + \|G - F\|^2$$
$$\geq \sigma^2 - \frac{\theta}{6} - 2\frac{\theta}{6} - 2\langle Z, F \rangle \geq \sigma^2 - \frac{\theta}{2} - 2\langle Z, F \rangle.$$

If further we have, $\|Z + G - F\|^2 \leq \sigma^2 - \theta$ then we have from above, $\langle F, Z \rangle \geq \frac{\theta}{4}$

Motivated by the above, we define the following 4 events, namely $\mathbf{E}_i, i = 1, \ldots, 4$

$$\mathbf{E}_1 := \left\{\|Z\|^2 \geq \sigma^2 - \frac{\theta}{6}\right\}, \; \mathbf{E}_2 := \left\{\langle Z, G \rangle \geq -\frac{\theta}{6}\right\}, \mathbf{E}_3 := \left\{\exists\, h_{\boldsymbol{w}_b, \boldsymbol{w}_t} \in \mathcal{H} \mid \hat{\mathcal{R}} \leq \sigma^2 - \theta\right\} \; \& \; \mathbf{E}_4 := \left\{\exists\, h_{\boldsymbol{w}_b, \boldsymbol{w}_t} \in \mathcal{H} \mid \langle F, Z \rangle \geq \frac{\theta}{4}\right\}$$

Thus our above argument can be summarized to say that if the events $\mathbf{E}_1$, $\mathbf{E}_2$ and $\mathbf{E}_3$ hold, then $\mathbf{E}_4$ will also hold. This we can write as, $\mathbb{P}(\mathbf{E}_1 \cap \mathbf{E}_2 \cap \mathbf{E}_3) \leq \mathbb{P}(\mathbf{E}_4)$. This implies, $\mathbb{P}(\mathbf{E}_4) \geq 1 - \mathbb{P}((\mathbf{E}_1 \cap \mathbf{E}_2 \cap \mathbf{E}_3)^c)$. But, by union bounding, $\mathbb{P}\left(\mathbf{E}_1^c \cup \mathbf{E}_2^c \cup \mathbf{E}_3^c\right) \leq \mathbb{P}(\mathbf{E}_1^c) + \mathbb{P}(\mathbf{E}_2^c) + \mathbb{P}(\mathbf{E}_3^c) \leq 3 - (\mathbb{P}(\mathbf{E}_1) + \mathbb{P}(\mathbf{E}_2) + \mathbb{P}(\mathbf{E}_3))$. Hence combining we have, $\mathbb{P}(\mathbf{E}_4) \geq -2 + (\mathbb{P}(\mathbf{E}_1) + \mathbb{P}(\mathbf{E}_2) + \mathbb{P}(\mathbf{E}_3))$

From equations 12 and 13 we obtain, that, $(1 - \mathbb{P}(\mathbf{E}_1)) \leq \exp\left(-\frac{n\theta^2}{288B^2}\right)$ and similarly for $(1 - \mathbb{P}(\mathbf{E}_2))$.

Thus substituting in above we get,

$$\mathbb{P}(\mathbf{E}_3) \leq 2\exp\left(-\frac{n\theta^2}{288B^2}\right) + \mathbb{P}(\mathbf{E}_4)$$

Thus we have proven what we had set out to prove, $\qquad\square$

## 6 Proof of the (Main)Theorem 4.1

A careful study of the proof of Lemma 5.4 would reveal that it can as well be invoked on $\mathcal{H}_\theta$.

And by doing so we get,

$$\mathbb{P}\left(\exists h_{(\boldsymbol{w}_{b,\frac{\theta}{2}},\boldsymbol{w}_{t,\frac{\theta}{2}})} \in \mathcal{H}_\theta \mid \frac{1}{n}\sum_{i=1}^n \left(y_i - h_{(\boldsymbol{w}_{b,\frac{\theta}{2}},\boldsymbol{w}_{t,\frac{\theta}{2}})}(\boldsymbol{s}_i,\boldsymbol{p}_i)\right)^2 \leq \sigma^2 - \theta\right)$$

$$\leq 2\exp\left(-\frac{n\theta^2}{288 \cdot B^2}\right) + \mathbb{P}\left(\exists h_{(\boldsymbol{w}_{b,\frac{\theta}{2}},\boldsymbol{w}_{t,\frac{\theta}{2}})} \in \mathcal{H}_\theta \mid \frac{1}{n}\sum_{i=1}^n h_{(\boldsymbol{w}_{b,\frac{\theta}{2}},\boldsymbol{w}_{t,\frac{\theta}{2}})}(\boldsymbol{s}_i,\boldsymbol{p}_i)z_i \geq \frac{\theta}{4}\right)$$

Using Lemma 5.3,

$$\leq 2\exp\left(-\frac{n\theta^2}{288 \cdot B^2}\right) + \frac{2^{2(d_B+d_T)+1}}{\theta^{(d_B+d_T)}} \cdot \left(W_B\sqrt{d_B}\right)^{d_B} \cdot \left(W_T\sqrt{d_T}\right)^{d_T} \cdot \exp\left(-\frac{2n\theta^2}{8^4 \cdot (B \cdot q\mathcal{C}^2)^2}\right)$$

$$+ 2\exp\left(-\frac{n\theta^2}{8^3 \cdot (B \cdot q \cdot \mathcal{C}^2)^2}\right)$$

$$(18)$$

Invoking Lemma 5.2 at $\theta = \frac{\epsilon}{q^2}$ and recalling that the $q \geq 1$ (by definition) we have,

$$\hat{\mathcal{R}}(h_{(\boldsymbol{w}_{b,\frac{\epsilon}{2\cdot q^2}},\boldsymbol{w}_{t,\frac{\epsilon}{2\cdot q^2}})}) \leq \hat{\mathcal{R}}(h_{(\boldsymbol{w}_b,\boldsymbol{w}_t)}) + \mathcal{C} \cdot J \cdot \epsilon \cdot \left(B + 2\cdot\mathcal{C}^2\right) \quad (19)$$

With respect to random sampling of the training data we define two events $\mathbf{E}_1$ (corresponding to the function class $\mathcal{H}$) and $\mathbf{E}_2$ (corresponding to the $\frac{\epsilon}{2q^2}$–cover of $\mathcal{H}$),

$$\mathbf{E}_1 := \left\{\exists h_{(\boldsymbol{w}_b,\boldsymbol{w}_t)} \in \mathcal{H} \mid \hat{\mathcal{R}}(h_{(\boldsymbol{w}_b,\boldsymbol{w}_t)}) \leq \sigma^2 - \epsilon - \mathcal{C}\cdot J\cdot\epsilon\cdot\left(B+2\cdot\mathcal{C}^2\right)\right\}$$

$$\mathbf{E}_2 := \left\{\exists h_{(\boldsymbol{w}_{b,\frac{\epsilon}{2\cdot q^2}},\boldsymbol{w}_{t,\frac{\epsilon}{2\cdot q^2}})} \in \mathcal{H}_\theta \mid \hat{\mathcal{R}}(h_{(\boldsymbol{w}_{b,\frac{\epsilon}{2\cdot q^2}},\boldsymbol{w}_{t,\frac{\epsilon}{2\cdot q^2}})}) \leq \sigma^2 - \epsilon\right\}$$

Thus if $\mathbf{E}_1$ is true, we can invoke the above inequality to get,

$$\hat{\mathcal{R}}(h_{(\boldsymbol{w}_{b,\frac{\epsilon}{2\cdot q^2}},\boldsymbol{w}_{t,\frac{\epsilon}{2\cdot q^2}})}) \leq \hat{\mathcal{R}}(h_{(\boldsymbol{w}_b,\boldsymbol{w}_t)}) + \mathcal{C}\cdot J\cdot\epsilon\cdot\left(B+2\cdot\mathcal{C}^2\right) \leq \sigma^2 - \epsilon - \mathcal{C}\cdot J\cdot\epsilon\cdot\left(B+2\cdot\mathcal{C}^2\right) + \mathcal{C}\cdot J\cdot\epsilon\cdot\left(B+2\cdot\mathcal{C}^2\right)$$

$$\leq \sigma^2 - \epsilon$$

Thus we observe that, $\mathbf{E}_1 \implies \mathbf{E}_2$ and thus $\mathbb{P}(\mathbf{E}_1) \leq \mathbb{P}(\mathbf{E}_2)$ we can invoke equation 18 to get,

$$\mathbb{P}\left(\exists h_{(\boldsymbol{w}_b,\boldsymbol{w}_t)} \in \mathcal{H} \mid \underbrace{\frac{1}{n}\sum_{i=1}^n (y_i - h_{\boldsymbol{w}_b,\boldsymbol{w}_t}(\boldsymbol{s}_i,\boldsymbol{p}_i))^2}_{\hat{\mathcal{R}}(h_{(\boldsymbol{w}_b,\boldsymbol{w}_t)})} \leq \sigma^2 - \epsilon\left(1 + \mathcal{C}\cdot J\cdot\left(B+2\cdot\mathcal{C}^2\right)\right)\right)$$

$$\leq \mathbb{P}\left(\exists h_{(\boldsymbol{w}_{b,\frac{\epsilon}{2\cdot q^2}},\boldsymbol{w}_{t,\frac{\epsilon}{2\cdot q^2}})} \in \mathcal{H}_{\frac{\epsilon}{q^2}} \mid \underbrace{\frac{1}{n}\sum_{i=1}^n \left(y_i - h_{(\boldsymbol{w}_{b,\frac{\epsilon}{2\cdot q^2}},\boldsymbol{w}_{t,\frac{\epsilon}{2\cdot q^2}})}(\boldsymbol{s}_i,\boldsymbol{p}_i)\right)^2}_{\hat{\mathcal{R}}(h_{(\boldsymbol{w}_{b,\frac{\epsilon}{2\cdot q^2}},\boldsymbol{w}_{t,\frac{\epsilon}{2\cdot q^2}})})} \leq \sigma^2 - \epsilon\right)$$

$$\leq 2\exp\left(-\frac{n\epsilon^2}{288 \cdot B^2 \cdot q^4}\right) + \frac{2^{2(d_B+d_T)+1}}{\left(\frac{\epsilon}{q^2}\right)^{(d_B+d_T)}} \cdot \left(W_B\sqrt{d_B}\right)^{d_B} \cdot \left(W_T\sqrt{d_T}\right)^{d_T} \cdot \exp\left(-\frac{2n\epsilon^2}{8^4 \cdot (B\cdot q\mathcal{C}^2)^2 \cdot q^4}\right) \quad (20)$$

$$+ 2\exp\left(-\frac{n\epsilon^2}{8^3 \cdot B^2 \cdot q^6 \cdot \mathcal{C}^4}\right)$$

Hence if the required probability has to be at least $1 - \delta$, it's necessary that we have,

$$(1 - \delta) \leq 2\exp\left(-\frac{n\epsilon^2}{288 \cdot B^2 \cdot q^4}\right) + \frac{2^{2(d_B+d_T)+1}}{\left(\frac{\epsilon}{q^2}\right)^{(d_B+d_T)}} \cdot \left(W_B\sqrt{d_B}\right)^{d_B} \cdot \left(W_T\sqrt{d_T}\right)^{d_T} \cdot \exp\left(-\frac{2n\epsilon^2}{8^4 \cdot B^2 \cdot q^6 \cdot \mathcal{C}^4}\right)$$
$$+ 2\exp\left(-\frac{n\epsilon^2}{8^3 \cdot B^2 \cdot q^6 \cdot \mathcal{C}^4}\right)$$

Note that, $\frac{n\epsilon^2}{8^3 \cdot B^2 \cdot q^6 \cdot \mathcal{C}^4} > \frac{2n\epsilon^2}{8^4 \cdot B^2 \cdot q^6 \cdot \mathcal{C}^4}$. Further recalling that $q, \mathcal{C} \geq 1$ we have, $\frac{n\epsilon^2}{288 \cdot B^2 \cdot q^4} > \frac{n\epsilon^2}{8^3 \cdot B^2 \cdot q^6 \cdot \mathcal{C}^4}$. Thus a necessary condition to satisfy the above inequality is obtained by weakening the above inequality by replacing all the three exponentials in there with the smallest of them,

$$1 - \delta \leq 2 \cdot \exp\left(\frac{-n\epsilon^2}{288 \cdot B^2 \cdot q^4}\right) \cdot \left(\left(\frac{4q^2}{\epsilon}\right)^{d_B+d_T} \cdot \left(W_B\sqrt{d_B}\right)^{d_B} \cdot \left(W_T\sqrt{d_T}\right)^{d_T} + 2\right)$$

And the necessary condition above leads to the lower bound,

$$q \geq n^{\frac{1}{4}} \cdot \left(\frac{\epsilon^2}{288 \cdot B^2} \cdot \frac{1}{\ln\left(\left(\frac{4q^2}{\epsilon}\right)^{d_B+d_T} \cdot \left(W_B\sqrt{d_B}\right)^{d_B} \cdot \left(W_T\sqrt{d_T}\right)^{d_T} + 2\right) + \ln\left(\frac{2}{1-\delta}\right)}\right)^{\frac{1}{4}} \tag{21}$$

The claimed lower bound in the theorem statement follows by further weakening the inequality above recalling that by definition we have, $q \leq \min\{d_B, d_T\}$.

## 7 The Experiment Set-up

In this section, we shall demonstrate that at a fixed number of total parameters, increasing the output dimension($q$) and increasing the training data as $q^2$ can cause a monotonic decrease in the training error of a DeepONet.

The advection-diffusion-reaction P.D.E. (Rahaman et al., 2022) (referred as the ADR PDE from here on) plays a crucial role in modeling various physical, chemical, and biological processes - and that shall be our example for the experimental studies. More specifically, given a function $f$ this PDE is specified as follows,

$$\frac{\partial u}{\partial t} = D\frac{\partial^2 u}{\partial x^2} + ku^2 + f(x), \quad x \in [0,1], t \in [0,1] \tag{22}$$

with zero initial and boundary conditions, and for our preliminary experiments we shall use $D = 0.01$ as the diffusion coefficient, and $k = 0.01$ as the reaction rate. We use DeepONets to learn the operator $\mathcal{G}$ mapping from $f(x)$ to the PDE solution $u(x,t)$. In this case the operator $\mathcal{G}_\theta$ will map the source terms $f(x)$ to the PDE solution $u(x,t)$. Given a choice of $m$ sensor points, we shall denote a discretization of $f$ onto the sensor points as the vector $\mathbf{f} \in \mathbb{R}^m$. Recalling the DeepONet operator loss, we realize that minimizing that is trying to induce, $\mathcal{G}_\theta(\mathbf{f}(x,t)) \approx \mathcal{G}(f(x,t))$.

For sampling $f$ we have considered the Gaussian random field(GRF) distribution. Here we have used the mean-zero GRF, $f \sim \mathcal{G}\left(0, k_l\left(x_1, x_2\right)\right)$ where the covariance kernel $k_l\left(x_1, x_2\right) = \exp\left(-\|x_1 - x_2\|^2/2l^2\right)$ is the radial-basis function (RBF) kernel with a length-scale parameter $l > 0$. For our experiments we have taken $l = 10^{-3}$. After sampling $f$ from the chosen function spaces, we solve the PDE by a second-order finite difference method to obtain the reference solutions.

For $n$ training data samples, the $\ell_2$ empirical loss being minimized is, $\hat{\mathcal{L}}_{\text{DeepONet}} \coloneqq \frac{1}{n}\sum_{i=1}^{n}\left(y_i - \mathcal{G}_\theta\left(f_i\right)\left(p_i\right)\right)^2$, where $p_i$ is a randomly sampled point in the $(x,t)$ space and $y_i$ is the approximate PDE solution at $p_i$ corresponding to $\boldsymbol{f}_i$ – which we recall was obtained from a conventional solver.

### 7.1 Implementations & Results

We created 10 DeepOnet models in each experimental setting such that each model has a depth of 5 and width varying between 24 and 50 for each layer while keeping the total number of training parameters approximately equal for each of those 10 models. For each case the branch input dimension is 40(i.e number of sensor points), and trunk input dimension is 2. The smallest number of training data $(n)$ we use is $10^4$ and twice we make a choice of 10 different $(q, n)$ values parameterizing the learning setups, once keeping the ratio $\frac{q}{\sqrt{n}}$ approximately constant and then holding the ratio $\frac{q}{n^{\frac{2}{3}}}$ almost fixed. All the DeepONet models were trained by the stochastic Adam optimizer at its default parameters.

The code for this experiment can be found in our GitHub repository (link).

**Experiments in the fixed $\frac{q}{\sqrt{n}}$ setting**. In this setting, the q value was varied from 5 to 50, in increments of 5. We have taken the starting value of $n$ as $10^4$. In Figure 2 we have plotted the training loss dynamics for these 10 models being trained over 120 epochs.

**Experiments in the fixed $\frac{q}{n^{\frac{2}{3}}}$ setting**. We repeat the above experiment but while appproximately fixing the value of $\frac{q}{n^{\frac{2}{3}}}$. The corresponding plots are shown in Figure 3.

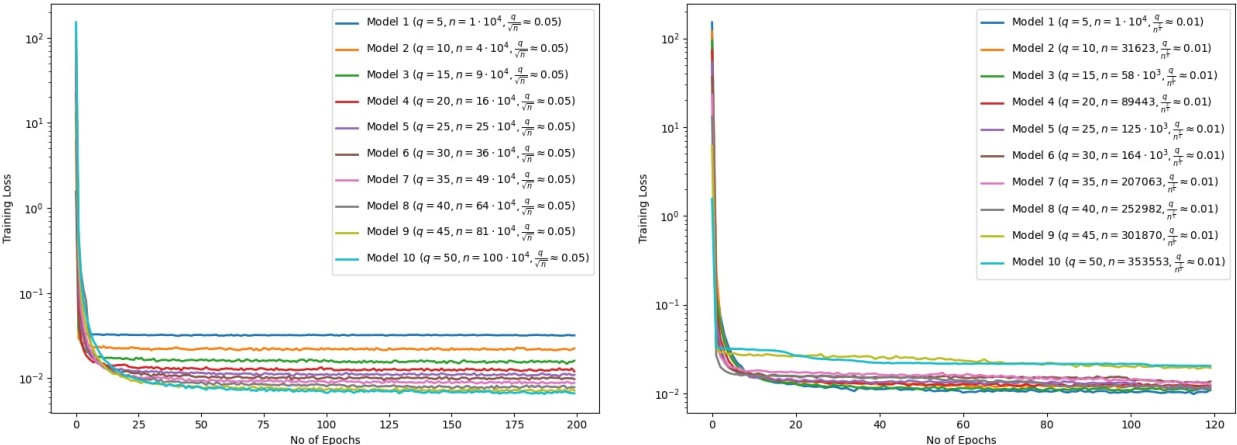

Figure 2: Training Loss vs Epoch in fixed $\frac{q}{\sqrt{n}}$ setting    Figure 3: Training Loss vs Epoch in fixed $\frac{q}{n^{\frac{2}{3}}}$ setting

Further experiments of the above kinds at other values of $D$ and $k$, for orders of magnitude above and below what's considered here, can be seen in Appendix A. In Appendix A.1 we shall show that the emergence of a scaling law as demonstrated for fixed $\frac{q}{\sqrt{n}}$ experiments persists even for fixed $\frac{q}{n^{\frac{1}{6}}}$ experiments - as is to be expected as the amount of available data increases for each model considered. A summary table of all parameters studied for the ADR PDE can be seen in Section A.2.

We draw two primary conclusions from the above results. *Firstly*, from Figure 2, we can observe that if $q$ and $n$ increase at a fixed $\frac{q}{\sqrt{n}}$ then performance increases almost monotonically. *Secondly*, from the Figure 3 it is clearly visible that the previous monotonicity is breaking - that is the rate of increase of data size in the later experiment was not sufficient to leverage the increase in the output dimension size of the branch and the trunk as was happening in the first figure.

## 8  Discussion

Our key result Theorem 4.1 shows that a certain data size dependent largeness of $q$ is needed if there has to exist a bounded weight DeepONet at that $q$ which can have their empirical error below the label noise threshold. From our experiments, we have shown that there is some non-trivial range of $q$ (the common output dimension) along which empirical risk improves with $q$ for a fixed model size - if the amount of training data is scaled quadratically with $q$. We envisage that trying to prove this "scaling law" can be a very interesting direction for future exploration in theory.

Secondly, we note that our result hasn't yet fully exploited the structure of the neural nets used in the branch and the trunk. Also, it would be interesting to understand how to tune the argument specifically for the different variations of this architecture (Kontolati et al., 2023), (Bonev et al., 2023) that are getting deployed, Lastly, we note that our result is currently agnostic to the PDE being attempted to be solved. There is a tantalizing possibility, that methods in this proof could be extended to derive bounds which can distinguish PDEs that are significantly hard for operator learning.

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

## A    Appendix

In this section we extended the scope of our demonstration by revisiting the experiments detailed in Section 7.1 and redoing them at further more values of the diffusion coefficient ($D$) and the reaction rate ($k$), as specified in Equation 22. This time we go a couple of orders of magnitude above as well as below the value of $D = k$ chosen in Section 7.1.

We conducted four sets of experiments, at different common values for $D$ and $k$, namely 1) 1 in Figure 4, 2) 0.1 in Figure 5, 3) 0.001 in Figure 6 and 4) 0.0001 in Figure 7.

Our findings indicate that for any given pair of $(D, k)$ value, at a fixed $\frac{q}{\sqrt{n}}$ setting, performance keeps on increasing monotonically i.e the best empirical loss obtained monotonically falls with increasing $q$. However, for the fixed $\frac{q}{n^{\frac{2}{3}}}$ setting, this monotonicity breaks, particularly for higher values of $D = k$.

Thus the key insights about a possible scaling law for DeepONets continues to hold as was motivated earlier in Section 7.1.

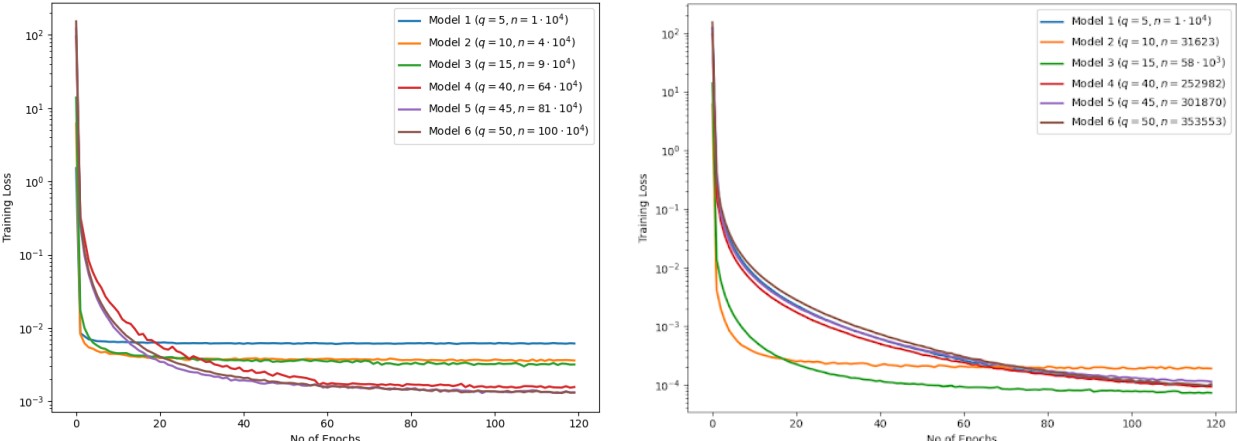

Figure 4: ($D$ & $k$ value as 1) Left: Training Loss vs Epoch in fixed $\frac{q}{\sqrt{n}}$ setting. Right: Training Loss vs Epoch in fixed $\frac{q}{n^{\frac{2}{3}}}$ setting.

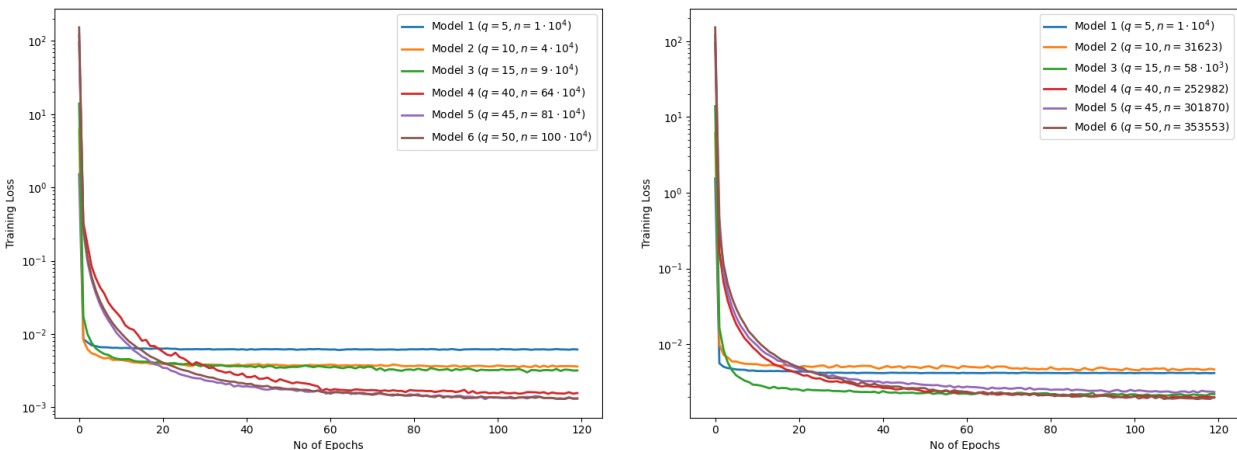

Figure 5: ($D$ & $k$ value as 0.1) Left: Training Loss vs Epoch in fixed $\frac{q}{\sqrt{n}}$ setting. Right: Training Loss vs Epoch in fixed $\frac{q}{n^{\frac{2}{3}}}$ setting.

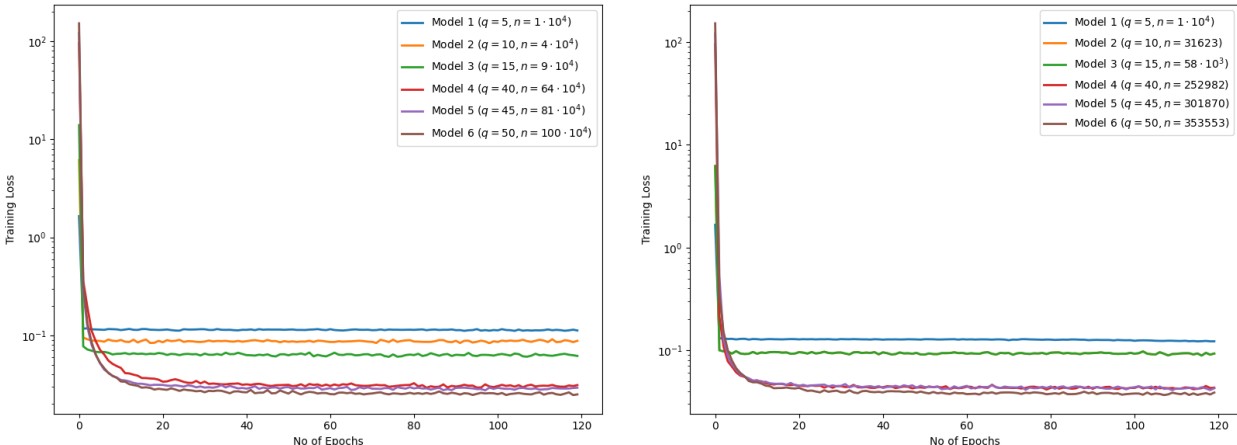

Figure 6: ($D$ & $k$ value as 0.001) Left: Training Loss vs Epoch in fixed $\frac{q}{\sqrt{n}}$ setting. Right: Training Loss vs Epoch in fixed $\frac{q}{n^{\frac{2}{3}}}$ setting

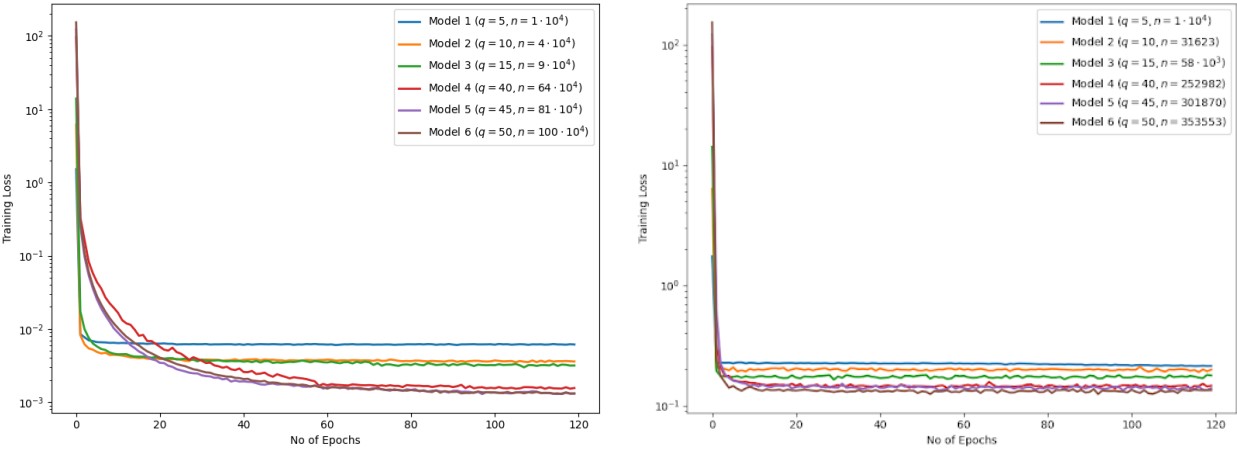

Figure 7: ($D$ & $k$ value as 0.0001) Left: Training Loss vs Epoch in fixed $\frac{q}{\sqrt{n}}$ setting. Right: Training Loss vs Epoch in fixed $\frac{q}{n^{\frac{2}{3}}}$ setting

## A.1   Experiment at a fixed $\frac{q}{n^{\frac{1}{6}}}$ setting on the ADR PDE

In here we conducted two sets of experiments, at $q$ and $n$ settings more closely inspired by Theorem 4.2 i.e. we chose a set of nets of increasing $q$ such that the size of the nets and $\frac{q}{n^{\frac{1}{6}}}$ are almost fixed. We do experiments at different common values for $D$ and $k$, namely 0.0001 & 1 in Figure 8.

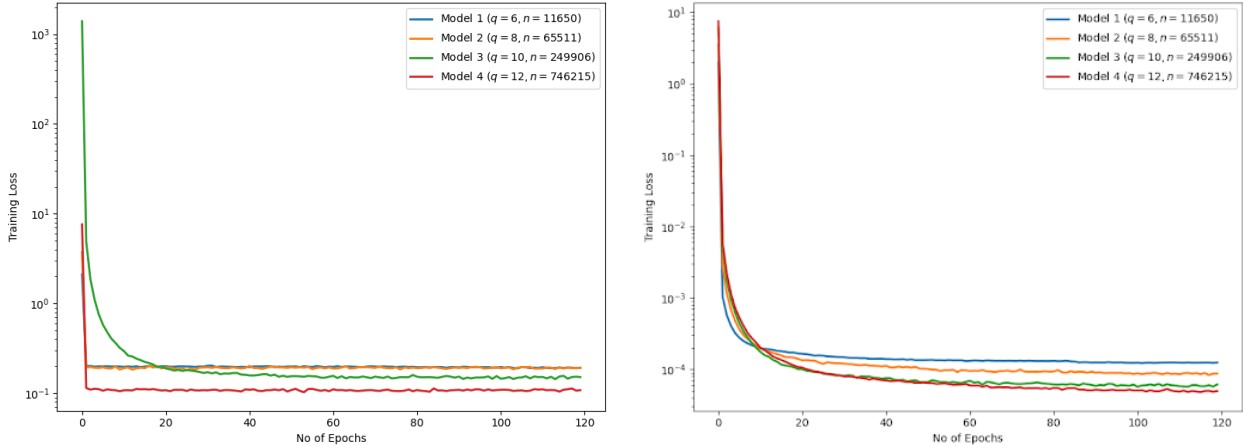

Figure 8: ($D$ & $k$ value as 0.0001) Left: Training Loss vs Epoch in fixed $\frac{q}{n^{\frac{1}{6}}}$ setting. ($D$ & $k$ value as 1)Right: Training Loss vs Epoch in fixed $\frac{q}{n^{\frac{1}{6}}}$ setting.

The above plots suggest that the monotonic performance improvement with increasing $q$ as seen in all earlier experiments continue to hold even at the scaling as chosen here.

## A.2   Table of Data for All Experiments on the ADR PDE

### A.2.1   Experiments where $\frac{q}{n^{\frac{1}{6}}}$ (and size of the operator net) are kept nearly constant

| # Trainable Parameters in the DeepONet | $q$ | $n, D = k = 10^{-4}$ | $n, D = k = 1$ |
|---|---|---|---|
| 18112 | 6 | 11650 | 11650 |
| 18316 | 8 | 65511 | 65511 |
| 18520 | 10 | 249906 | 249906 |
| 18724 | 12 | 746215 | 746215 |

Table 1: The last 2 columns correspond to Figure 8

### A.2.2   Experiments where $\frac{q}{n^{\frac{1}{2}}}$ (and size of the operator net) are kept nearly constant

| # Trainable Parameters in the DeepONet | $q$ | $n, D = k = 10^{-4}$ | $n, D = k = 10^{-3}$ | $n, D = k = 10^{-2}$ | $n, D = k = 10^{-1}$ | $n, D = k = 1$ |
|---|---|---|---|---|---|---|
| 18010 | 5 | 10000 | 10000 | 10000 | 10000 | 10000 |
| 18520 | 10 | 40000 | 40000 | 40000 | 40000 | 40000 |
| 18568 | 15 | 90000 | 90000 | 90000 | 90000 | 90000 |
| 18719 | 40 | 640000 | 640000 | 640000 | 640000 | 640000 |
| 18714 | 45 | 810000 | 810000 | 810000 | 810000 | 810000 |
| 18760 | 50 | 1000000 | 1000000 | 1000000 | 1000000 | 1000000 |

Table 2: The column for $D = k = 10^{-2}$ corresponds to Figure 2 and the rest of the last 5 columns, starting from the rightmost above, correspond to the left column of Figures 4, 5, 6, and 7

### A.2.3 Experiments where $\frac{q}{n^{\frac{2}{3}}}$ (and size of the operator net) are kept nearly constant

| #Trainable Parameters in the DeepONet | $q$ | $n, D = k = 10^{-4}$ | $n, D = k = 10^{-3}$ | $n, D = k = 10^{-2}$ | $n, D = k = 10^{-1}$ | $n, D = k = 1$ |
|---|---|---|---|---|---|---|
| 18010 | 5 | 10000 | 10000 | 10000 | 10000 | 10000 |
| 18520 | 10 | 31623 | 31623 | 31623 | 31623 | 31623 |
| 18568 | 15 | 58000 | 58000 | 58000 | 58000 | 58000 |
| 18719 | 40 | 252982 | 252982 | 252982 | 252982 | 252982 |
| 18714 | 45 | 301870 | 301870 | 301870 | 301870 | 301870 |
| 18760 | 50 | 353553 | 353553 | 353553 | 353553 | 353553 |

Table 3: The column for $D = k = 10^{-2}$ corresponds to Figure 3 and the rest of the last 5 columns, starting from the rightmost above, correspond to the right column of Figures 4, 5, 6, and 7

