# OpenReview forum: "Size Lowerbounds for Deep Operator Networks"
_TMLR — Accepted by TMLR_

### Review · Reviewer_mYsr · 2023-09-03

**Summary Of Contributions:**

This paper studies the necessary lower bound on the size of DeepONet required to attain a given empirical training error. It is shown that the output dimension of the DeepONet needs to scale $O(\sqrt{n})$ where $n$ denotes the number of training data when the training error is achieved below a label noise dependent threshold. Some numerical experiments are provided to verify the necessary growth of output dimension as a function of the size of training data to monotonically decrease the training error.

**Audience:**

Yes

**Broader Impact Concerns:**

No.

**Claims And Evidence:**

Yes

**Requested Changes:**

- One major concern I have is about the statements of the main results. For example, in Theorem 4.1, I am confused by which parameter is fixed (and which needs to be bounded). If I understand it correctly, given a fixed sample size $n$ the goal is to obtain a lower bound on the output dim $q$ as a function of $n$. However, the introduction of parameter $\theta$  complicates the process and makes me really confused. In the preample of Theorem 4.1, $\theta \leq \epsilon/q^2$ and $n \geq 288B^2/\theta^2 \ln{4/(1-\delta)}$. This means that $n \gtrsim q^4/\epsilon^2$ or equivalently $q \lesssim n^{1/4}$. Would not that contradict with the lower bound requirement $q\gtrsim O(n^{1/2})$? The author should elaborate and clarify this in the revision.

- I strongly recommend the authors elaborate more on the Setup, especially the discussion on the training datasets. For readers who do not know the background of operator learning, the current description of training dataset is way too sketchy. If I understand it correctly, the data set should be the input and output functions. But what is the meaning of $y_i$?

-In Theorem 2.1 and several other places, the authors mention that $p$ is the size of the trunk net. Is this the total number of neurons or just the width of the network?

The authors are recommended to submit a major revision of the paper addressing the comments above before it is considered being published in TMLR.

**Strengths And Weaknesses:**

- Strengths. Proving complexity lower bounds of DeepONet is an important question for operator learning while has less been explored. The paper proved an interesting  lower bound for learning a generic operator with DeepONet. To the best of my knowledge, this result seems to be new and relies on the recent result on robustness of neural networks by  Bubeck \& Selker 2023.

- Weaknesses. I do not have much to comment on the weakness, but I do have some concerns and questions on the statements of main results, which can lead to a major mistake/weakness of the paper. See the Questions below.

---

> ### Author Response · Authors · 2023-09-24
> **We have made the necessary edits in the writing to address all the issues raised**
>
> Thank you for your very careful reading of the draft.
>
> In the writing below we have explained how we have made edits in the draft to address all the concerns raised by you.
>
> > "...in Theorem 4.1, I am confused by which parameter is fixed (and which needs to be bounded). If I understand it correctly, given a fixed sample size $n$ the goal is to obtain a lower bound on the output $\operatorname{dim} q$ as a function of $n$. However, the introduction of parameter $\theta$ complicates the process and makes me really confused..."
>
> We are very grateful to the reviewer for pointing out this error that had occurred in our writing. Kindly see how we have changed the theorem statement in the revision that we have uploaded. Note that in this new version of Theorem 4.1 (a) it is now explicit that the lowerbound on the common output dimension of the branch and the trunk is ${\rm (training-data-size)}^{1/6}$ and (b) now the result holds for any value of the training data size and (c) the variable theta is not needed anymore in writing this statement.
>
> We would like to clarify that compared to the original submission, in the revision nothing had to be fundamentally changed in the core calculation that happens in the proof of Theorem 4.1 (and the associated lemmas). We have merely restructured the proof and simplified certain steps to make the theorem more transparent - and fixed the error in our writing that had happened in the associated commentary.
>
> In light of this new version of Theorem 4.1, the experiment can be seen as motivating that to get monotonic improvement of performance with an increase in training data, the output layer size of the nets might have to grow at a stronger rate with data size -- than what the theorem suggests as being necessary to lower empirical error below the noise threshold. Thus a very interesting direction of future pursuit is motivated.
>
> >"I strongly recommend the authors elaborate more on the Setup, especially the discussion on the training datasets. For readers who do not know the background of operator learning, the current description of training dataset is way too sketchy. If I understand it correctly, the data set should be the input and output functions. But what is the meaning of $y_i$ ?"
>
> In this work, the loss function for DeepONets that we study is training this architecture in the supervised setting. Hence $y_i$ are the labels corresponding to the input data $(\vec{s}_i,\vec{p}_i)$. In a typical use-case of this, $y_i$ would be the evaluation of the true PDE solution at the point $\vec{p}_i$ when the PDE's inhomogeneous term/the forcing function discretizes to $\vec{s}_i$ on some fixed grid.
>
> Kindly see the new segment that we have added at the end of Section 3, titled ``An Example of a DeepONet Loss'' - where we have demonstrated how a DeepONet loss function is setup and thus put the setup of that section in context, using an explicit example of wanting to solve in one-shot the forced pendulum's differential equations at arbitrary forcing functions.
>
> Of course we note that the lower bound we have derived is universal and does not depend on the specific differential equation system being targeted.
>
> > "In Theorem 2.1 and several other places, the authors mention that $p$ is the size of the trunk net. Is this the total number of neurons or just the width of the network?"
>
> Kindly note that now we have uniformized the notation so that the variable $q$ has the same meaning as in review of Theorem 2.1 and our result Theorem 4.2. In both the cases, now $q$ denotes the common output dimension of the trunk and the branch.

---

### Review · Reviewer_pViX · 2023-09-08

**Summary Of Contributions:**

The authors propose a lower bound on the size of DeepONets (meaning the number of trunk and branch nets), mainly as a function of the number of samples in the training data set $n$, with a resulting lower bound $\Omega(\sqrt{n})$. Inspiration is drawn from the work of Bubeck and Sellke (2023). The theory is supplemented by an experiment where DeepONets were trained for fixed ratios of $q/n^\gamma$, suggesting a sort of scaling law.

**Audience:**

Yes

**Broader Impact Concerns:**

/

**Claims And Evidence:**

No

**Requested Changes:**

It would be great if the authors can address or clarify the weaknesses mentioned above. Some additional remarks:
- RC1) Regarding W1: I would suggest to clarify further that the obtained lower bounds are more of an artefact from the way the training data is constructed, rather than an intrinsic property of DeepONets or the underlying PDE (as was the case in Theorem 2.1).
- RC2) Regarding W4: I would suggest to clarify how Theorem 4.1 addresses the second weakness mentioned after Theorem 2.1.
- RC3) In Lemma 5.1 one uses the notation $N$ before defining it at the end of the lemma, which is very confusing. Similar in Lemma 5.2 with $\hat{\mathcal{R}}$.
- RC4) In the beginning of page 10 (proof of Theorem 4.1) it is written that one requires that the probability is at least $1-\delta$. In the proof of the main result of Bubeck and Sellke (2023) one requires that the probability is at most $\delta$. Why is it different here?
- RC5) Experiments could be repeated with different constants, and perhaps different PDEs.

**Strengths And Weaknesses:**

*Strengths*

- S1) Theory for operator learning has mainly focused on providing upper bounds on the network size in terms of the error tolerance. It is a very interesting topic to investigate whether corresponding lower bounds can also be proved, and how these depend on all parameters. This work could be a valuable contribution in this area.
- S2) Although the evidence is rather limited, the scaling law that the experiment shows is very interesting and would be useful for practitioners.

*Weaknesses*
- W1) Just like the results of Bubeck and Sellke (2023), the lower bounds are only relevant in the case of noisy training data. In the context of DeepONets, noise in the training data can occur when physical measurements are used, or (more commonly) when the training data set is obtained through a numerical solver. Suppose one wants to guarantee an accuracy of $\sigma^2/2$ in the error bound of Theorem 4.1 (bottom of page 6), then one can either (1) use a large enough DeepONet following Theorem 4.1 or (2) one can simply use a slightly better numerical solver such that the variance of the noise is $\sigma^2/2$ rather than $\sigma^2$. I believe that such considerations are important and put the proposed results in a different context, as results such as Theorem 2.1 do not depend on data noise.
- W2) Notation throughout the paper is very inconsistent and therefore hard to follow. The true operator is denoted by both $\mathcal{G}$ and $g$, the input function of the DeepONet is denoted by either $f$ or $s$, whereas $s$ is at the same time the total number of parameters in the DeepONet. The number of branch/trunk nets is denoted by both $p$ and $q$.
- W3) There sometimes is a lack of mathematical rigour, which again makes the paper harder to follow. For instance, in Definition 1 it is unclear where the $(y_i, (s_i,p_i))$ is sampled from (which space, which distribution). In Section 1.1 one considers a compact domain $D$ yet in Definition 2 the trunk and branch nets are considered over $\mathbb{R}^{d_i}$. It is unclear which setting is considered in the main results. In Definition 3, which vector norm is used? In Definition 4, over which set is the supremum taken? In the main theorem, $\theta$ is defined in terms of $q$ but $q$ is undefined at that point and then only later a lower bound for $q$ is introduced.
- W4) The authors state that the downside of Theorem 2.1 is that it doesn't make a connection between the lower bound and the size of the branch net, but also main theorem only gives a lower bound on the number of trunk nets, and not (at least not explicitly) on the size of the branch net.
- W5) Finally, the main result is stated in a way that is very hard to interpret. In particular, there are three equations that link $q$, $n$ and $\epsilon$: namely $\theta \leq \epsilon/q^2$, the assumption on the training set size $n\geq C_1 / \theta^2$ and the final lower bound $q\geq C_2 \theta \sqrt{n}$ (simplified). Rearranging these equations leads to $q^2 \lesssim \epsilon n$. This can either be interpreted as  (1) an upper bound on $q$ that needs to hold if the lower bound is valid (which is a bit weird as the goal is to obtain a *lower* bound), or (2) a lower bound for $\epsilon$ (which is hopefully smaller than $(1+\mathcal{C} J(B+2\mathcal{C}^2)^{-1}$ as otherwise the assumption of the theorem is invalid). Finally, the total number of parameters of the DeepONet $d_B$ and $d_T$ (and their sum $s$) also depend on $q$. This means that the current upper bound on $q$ is more of an implicit relationship between various quantities, rather than an explicit upper bound. All of this makes it currently very hard to get a clean understanding of the meaning of the main result.

---

> ### Author Response · Authors · 2023-09-24
> **We have added more experiments and made changes to the writing to clarify the issues raised.**
>
> Thank you for your very careful reading of the draft.
>
> In the writing below we have explained how we have made edits in the draft to address all the concerns raised by you.
>
> Firstly, we note that your observation is entirely correct that ``noise in the training data can occur when physical measurements are used, or (more commonly) when the training data set is obtained through a numerical solver.'' Hence ours can be seen as a proof of the required size of DeepONets to be able to lower empirical error in the presence of epistemic uncertainity.
>
> Secondly, we have now edited the statements of Theorem 4.1 and Theorem 4.2 to clarify the issues that were pointed out. Note that these updated theorems now apply to any size of the training data. Thirdly, we have now edited Definition 1 and 4 to clarify that the data is from certain compact domains - though in the other definitions which pertain to the general function class we have kept the domains to be the full Euclidean spaces. Fourthly, the notation in equation 1 has also been simplified for clarity.
>
> Lastly, we hope that it would be clear from context that the notation for $s$ for size of the DeepONet that is used only in the statement of Theorem 4.2 is different from the $\vec{s}$ which is the input to the branch net.
>
> >RC1) Regarding W1: I would suggest to clarify further that the obtained lower bounds are more of an artefact from the way the training data is constructed, rather than an intrinsic property of DeepONets or the underlying PDE (as was the case in Theorem 2.1)
>
> In the comparison given below our review of Theorem 2.1 (prior work) we have now emphasized that this prior work specific to Burgers' PDE did not account for having noise in the labels while our lowerbound (Theorem 4.2) is not only independent of the PDE being solved but is also specifically targeted to understand the architectural constraint on DeepONets arising from noisy labels.
>
> > RC2) Regarding W4: I would suggest to clarify how Theorem 4.1 addresses the second weakness mentioned after Theorem 2.1.
>
> Kindly see the updated comparisons given below Theorem 2.1, where all issues have been made clearer.
>
> Now we have also uniformized the notation so that the variable $q$ has the same meaning as in review of Theorem 2.1 and our result Theorem 4.2.
>
> > RC3) In Lemma 5.1 one uses the notation N before defining it at the end of the lemma, which is very confusing. Similar in Lemma 5.2 with $\hat{\mathcal R}$
>
> Kindly see the updated statements of Lemma 5.1 and 5.2 where we have now fixed this issue with our writing.
>
> >RC4) In the beginning of page 10 (proof of Theorem 4.1) it is written that one requires that the probability is at least $1-\delta$. In the proof of the main result of Bubeck and Sellke (2023) one requires that the probability is at most $\delta$. Why is it different here?
>
> May we kindly draw your attention to the statement of Theorem 3 (end of page 7) of the Neurips version of the paper you have referred to, \url{https://proceedings.neurips.cc/paper/2021/file/f197002b9a0853eca5e046d9ca4663d5-Paper.pdf}. Then it becomes clear that the meaning of the variable $\delta$ is the same between there and our Theorem 4.2
>
> >RC5) Experiments could be repeated with different constants, and perhaps different PDEs.
>
> Kindly see the newly added Appendix A where we have extended the existing experiments to explore values of the coefficients D and k in the current PDE which are two orders of magnitude below and above the current setting. One can see that the "scaling law" that was indicated in the prior experiments continue to hold.

---

> > ### Comment · Reviewer_pViX · 2024-01-22
> > **Reply to authors**
> >
> > Thank you for making this comment visible to me. I thank the authors for addressing my remarks, improving the exposition of their results and adding more experiments.
> >
> > In my opinion, it is still not possible to interpret the main result Theorem 4.1 in a clear way, due to the following reasons:
> > * The lower bound of equation (2) still depends on $d_B$ and $d_T$, which very much depend on $q$ through Definition 2. This seems to have rather incomprehensible consequences: suppose we substantially increase $n$, then (2) tells us that $q$ needs to be substantially higher as well. As a result, $d_B$ and $d_T$ must increase as well. However, when $d_B$ and $d_T$ increase, the RHS of (2) decreases such that now $q$ can be lower as well. This seems to jumpstart a never-ending loop. I wonder whether this is related to the upper bound $q\leq Q$ (with $Q$ fixed) that the authors had in the previous version, that they have now replaced by $Q:=\max(d_B, d_T)$. I would greatly appreciate a clear interpretation of this issue.
> > * Regarding RC4, let me rephrase my question. In the current version of Theorem 4.1 a factor $\ln(2/(1-\delta))$ appears in (2) (with probability $1-\delta$ wrt sampling of the data), whereas in Theorem 3 of Bubeck and Sellke (2023) a factor $\ln(4/\delta)$ appears. Note the $1-\delta$ vs. $\delta$. In Bubeck and Sellke (2023) I interpret this as follows: if you want to make a statement with a high probability, then the statement will be less strong. In Theorem 4.1 this is somehow the opposite, so I would be thankful for an explanation for this. In addition, I would advise to rewrite Theorem 4.1 more in the form of Theorem 3 of Bubeck and Sellke (2023) in case they are indeed of a similar form. Now my initial interpretation was not the same; it was not clear to me that (2) only holds probabilistically.
> > * In the proof of Theorem 4.1, it is unclear to me how the authors go from equation (20) to (21). I will refer to the two equations in between by (20+) and (20++), let A be the RHS of (20+) and B the LHS of (20++). Below (20) it is stated that it is necessary to have that $1-\delta \leq A$. Then, $B$ is defined for which it holds that $A\leq 2B$. The authors then suggest that $1-\delta \leq 2B$ implies that $1-\delta \leq A$ is satisfied. This would have only been the case if $A\geq 2B$ as then you have $1-\delta \leq 2B \leq A$. Am I missing something there?
> >
> > I thank the authors in advance for their further consideration of my remarks.

---

> ### Author Response · Authors · 2024-01-23
> **Further Clarifications (And The Submission Has Been Updated to Reflect Them)**
>
> Thanks a lot for your detailed reading and your insightful comments.
>
> Please see the following clarifications to the 3 points you have raised,
>
> -  Kindly note that the *training data size $n$ and the DeepONet class $\cal H$ is fixed before the theorem is invoked* and hence the parameters $d_B$, $d_T$, $W_B$ and $W_T$ and $q$ (and also the input dimensions $d_1$ and $d_2$) have been fixed right at first. Note that any such choice of parameters allows for a large number of architectures in the class of DeepONets $\cal H$
>
>    -- and we are asking if any member of this class satisfies the condition that w.h.p over sampling data, its empirical error is below a certain threshold determined by the noise in the labels.
>
>
>    Our Theorem 4.1 answers that the above demand is not possible to meet unless to start off $q$ was somewhat large as given by the lowerbound -- which depends only on the hyperparameters fixed by the choice of the function class.
>
>
> - Kindly note that as $1-\delta$ gets larger/closer to $1$ our probability requirement is getting stronger and then $\ln(\frac{2}{(1-\delta)})$ term is getting smaller and since this occurs in the denominator of the RHS in equations $2$ and $3$, it is making the lowerbound larger.
>
>    So $1-\delta$ getting close to $1$ makes the restriction on $q$ stronger - and we posit that this matches the natural intuition that architectural constraints should get stronger when asking for the good empirical error event to happen with high probability.
>
>   We think that a very direct comparison to the setup of Theorem 3 of Bubeck and Sellke (2023) is hard to do because in their setting $\delta$ going to $0$ (and hence asking for higher guarantees) will also push up the required values of $n\epsilon^2$ and then there is combined impact of all these on the effective lowerbound on the Lipschitz constant that they are after.
>
> - We take your point that we had an ambiguity in the writing between equations $20$ and $21$. Please see the refined explanations that we have given
>
>   -- and which now infact leads to a better bound than the earlier version.
>
>   Please note that since we are looking for a lowerbound, we need to get to a necessary (and not a sufficient) condition from equation $20$ and hence we weaken the bound from the equation that you called $20+$ to what you called $20++$.

---

> > ### Comment · Reviewer_pViX · 2024-01-24
> > **Reply to the authors**
> >
> > Many thanks for the quick reply.
> >
> > Regarding point 1: thanks for clarifying the main result. Given my initial confusion, I believe that there still is room for improvement in terms of how the theorem is presented to allow for an easier interpretation. Some additional thoughts:
> > - If I understand it correctly, then (2) is a *necessary* condition for a member of the hypothesis class to have a small empirical error (whp). Is it also a *sufficient* condition? If yes, it would be useful to rephrase the theorem in the form "if (2) holds, then the empirical error is ...". If not, it would be very interesting for the readers if you could comment on which other conditions should be met for the empirical error to be small.
> > - Am I correct that the practical use of Theorem 4.1 is as follows? The practitioners fix all parameters and verify whether (2) holds -- if it doesn't hold then there is no hope for a guaranteed smallness of the empirical error. Is there a way to get a more direct recommendation for practitioners? I.e. given a certain error tolerance and data set size, the network should be at least be of a certain size. Additionally, an example with concrete numbers for all size parameters could be helpful.
> > - The authors seem to imply that my example for increasing $n$, leading to first increasing and then decreasing $q$, is not immediately relevant as all parameters (including $n$) are fixed. However, in multiple locations in the paper they claim that $q$ should scale as $\Omega(n^{1/4})$, e.g. in the abstract. I do not understand how this setting is different from my example, nor do I see why this scaling follows immediately from (2) because of the implicit relation between $q$ and $d_B$ and $d_T$.
> >
> > Regarding point 3: I have revisited the math around equation (20) and do not seem to grasp one crucial point. I believe the authors want to prove that $P(\text{smallness of empirical error})=P(E_1)$ is higher than $1-\delta$, as written in Theorem 4.1. Yet, in equation (20) the authors prove an inequality in the form $P(E_1) \leq \ldots$, implying that the authors are proving a condition for $P(\text{smallness of empirical error})=P(E_1)$ to be *lower* than $1-\delta$. Am I missing something here?

---

> ### Author Response · Authors · 2024-01-26
> **Further Explanations**
>
> We thank the reviewer for these detailed questions.
>
> Let us clarify the issues raised as follows,
>
> - The question of sufficient condition on DeepONets to have low empirical error is a very interesting one and quite the opposite direction to our work here which focusses on the necessary conditions.
>
>    We believe that existing results on DeepONets can be carefully threaded together to get to a sufficient condition and we can outline a possible proof as follows : firstly we recognize that sufficient condition for the existence of small DeepONets with low population risk are known (and they are essentially proofs of universal approximation for this architecture) as in this paper, https://doi.org/10.1016/j.neunet.2022.06.019 and strong bounds (based on Rademacher complexity) are known for the gap between empirical and population loss of such architectures, as in this paper https://arxiv.org/abs/2205.11359. It is plausible to surmise that a careful merger of such pairs of results can result in sufficient conditions for constructing DeepONets with low empirical error.
>
> -  Please note that the experimental studies that we give essentially indicate the presence of a scaling law for DeepONets i.e by how much must the common output dimension of the branch and the trunk net be increased so that availability of extra data can be leveraged to an advantage assuming the total number of trainable parameters cant be increased.
>
>    This observation that we have demonstrated through careful experiments can be seen as an experimental setup recommendation that an empiricist can take away from our work.
>
>     And this experimental study is not proven by the theorem we prove but is only motivated by it.
>
> - Kindly note that equation 20 proves an upperbound on the good event that there exists a DeepONet in the class $\cal H$ with low empirical error.  But what we want is the criteria for this good event to happen with high probability i.e $1 - \delta$. So this creates tension - and this tension is the source of the restriction that we derive on the function class.
>
>
>   That is our proof technique. Please note that equation $20+$ could have very well been our final answer in the theorem. It's purely for reasons of easy interpretability that we weaken the condition in equation $20+$ to get to the necessary condition as given in $21$.

---

> > ### Comment · Reviewer_pViX · 2024-01-30
> > **Reply to the authors**
> >
> > Thank you for your comments.
> >
> > > Kindly note that equation 20 proves an upperbound on the good event that there exists a DeepONet in the class with low empirical error. But what we want is the criteria for this good event to happen with high probability i.e. $1-\delta$
> >
> > I fully agree with these statements, but don't understand how you can achieve the latter by providing an *upper* bound on the probability of that "good event". The theorem implies that a *lower bound* is needed, as the statement mentions a "probability of *at least* $1-\delta$". From my understanding of the proof of Theorem 4.1, this is not what is done. Instead, an *upper bound* is proved and $q$ is chosen such that this upper bound is larger than $1-\delta$. This is very different from proving a lower bound and does not imply the statement of the theorem. Many thanks in advance for clarifying this point.

---

> ### Author Response · Authors · 2024-01-30
> **About Theorem 4.1's Proof**
>
> What we want is this : $1 - \delta \leq P [ {\rm low-error-DeepONet-exists}]$
>
> Equation $20$ shows that there is a function of $q$ (say $f(q)$) s.t $P [ {\rm low-error-DeepONet-exists}] \leq f(q)$.
>
> And this upper-bound holds unconditionally on $q$.
>
> Hence $1 - \delta \leq P [ {\rm low-error-DeepONet-exists}]  \implies 1 - \delta \leq f(q)$
>
> In other words, a necessary condition for $1 - \delta \leq P [ {\rm low-error-DeepONet-exists}]$,
>  is that $q$ be restricted as $1 - \delta \leq f(q)$.
>
> To emphasize again this is not a proof of sufficient condition on the architecture for the DeepONet to have low errors.

---

> > ### Comment · Reviewer_pViX · 2024-01-30
> > **Reply to the authors**
> >
> > Many thanks for your patience in clarifying this point. I would suggest to remove the word 'ensure' from Theorem 4.1 and 4.2; this was the source of my confusion as it can lead to the main result being interpreted as a sufficient condition.

---

> ### Author Response · Authors · 2024-01-31
> **The Language in the Theorem Statements Have Been Edited**
>
> Thanks for your comments!
>
> Kindly see the updated draft where we have edited the language of the statements in Theorems 4.1 and 4.2 to be more clear. We have also added a clarifying line in the paragraph below Theorem 4.1, making it more explicit that we are proving a necessary condition for the existence of good DeepONets of a certain type.

---

### Review · Reviewer_qWBR · 2023-10-03

**Summary Of Contributions:**

This paper analyses Deep Operator Networks and gives a bound which relates the choice of the $q$ parameter (the dimension of the latent space connecting the 'branch' and 'trunk' networks in a DeepONet ) to the size $n$ of the empirical data set. To my knowledge, the relationship derived is novel and of interest to the operator learning literature for PDE surrogate modelling.

From my understanding, I see two main contributions of this paper.
1) How to derive a bound which relates the choice of $q$ with the choice of $n$; and,
2) Empirical evidence to show that monotonic convergence can be achieved by keeping $q/\sqrt{n}$ constant with a fixed parameter budget on a certain PDE.

For the first point, deriving this relationship is the main contribution of the paper, with Lemma 5.1-5.4 necessary to pave the path for achieving the bound given in Theorem 4.1. As the authors note, there is a discrepancy between the bound they derive and what achieves monotonic convergence in their experiments "Later, in Section 8, we shall conduct an experimental study motivated by the above and reveal something more than what the above theorem guarantees.". From this, I see this paper as a first step towards quantifying an optimal scaling relationship between $q$ and $n$ in DeepONets, but there is still future work to possibly sharpen this result. Because of this, I find what I consider their second contribution to be rather interesting and understated compared to point 1) . Motivated by the relationship they derive, the authors conduct a numerical convergence test and identify monotonic convergence at a rate different to what they found in Theorem 4.1. Whilst welcomed, I view this result as separate to their theoretical analysis and possibly suggestive of a stronger result yet to be proved.

**Audience:**

Yes

**Broader Impact Concerns:**

-

**Claims And Evidence:**

Yes

**Requested Changes:**

Suggested changes:

I might be incorrect in my understanding here, but currently I see that the experiments run show a sharper rate than what is proven in the main theorem. If this is the case, I believe the text could be improved with the addition of a simple table of $q$ and $n$ comparing the values used in the experiments and the rate required from the theory to make this clear. I also think having a small example of the rate predicted by the theory, even if it is a weaker experiment, would help in the presentation of the results. Currently it is a little difficult to immediately understand how the current bound influences the choice of $q$ and $n$ in the experiments.

I also recommend reordering some of the sections to improve the flow of the paper. I currently do not understand why the proofs of the lemmas 5.1-5.4 are listed in section 7 when section 5 states all of the lemmas and section 6 begins with:
"A careful study of the proof of Lemma 5.4 would reveal that it can as well be invoked on Hθ.".
For the readability of the document, I would recommend just putting the proofs of the lemmas with their statements in section 5, especially seeing as the current section 6 is also proof based and follows on from the proofs of the lemmas.

Minor comments:

The authors swap between PDE, PDEs and P.D.E in various sections (see Introduction and Section 8). This type of inconsistency is present throughout the document, for instance the authors swap between Physics Informed Neural Networks and Physics Inspired Neural Networks in an introductory paragraph. Making sure that abbreviations are consistent would aide the reader throughout the document.

There are several spelling/grammar mistakes throughout the current document which can be straightforwardly fixed, the ones I can easily garner are: "oftenn" and "reviewed" (instead of review I think) in the second paragraph of the introduction; Equation 1 and the equation at the beginning of Section 1 are missing punctuation; Definitions 1 and 3 are not complete sentences. I would request that the authors ensure that the sentences are complete, especially around the mathematics and definitions, to help the flow of the document.

**Strengths And Weaknesses:**

Strengths:
The proofs in the paper is cleanly set out and reasonable to follow. Splitting Theorem 4.1 into four smaller steps makes it easier to identify how the relationship between $q$ and the dataset size is derived, and the splitting of events $E_i$ make this proof well structured. The paper also clearly lays out previous work and restates relevant theorems pertaining to the approximation theory of DeepONets. This paper is strong in completing its main objective of identifying how to connect the parameter $q$ with the dataset size $n$.

Weaknesses:
The computational study in this paper is limited compared to the theoretical analysis. Whilst the authors present two settings, one for the case $q/\sqrt{n}$ is fixed and one whilst $q/n^{2/3}$ is fixed, neither of these correspond to the rate that they derive in their analysis. Whilst, from what I gather, their computational evidence here is stronger than what they prove, for completeness the paper should have a small experiment for the rate they derive. The authors also only consider one test problem of a single PDE, with variations of the coefficients in the appendix. Whilst a computational study is not the focal point of the paper, it would be interesting to know if this monotonic convergence for $q/\sqrt{n}$ occurs elsewhere, or if this is just relevant to the PDE the authors consider. Currently, it is unclear to me whether the authors are trying to convey a conjecture that the rate $q/\sqrt{n}$ might be expected in practice generally or if this rate could be given from specific additional structure of the particular PDE investigated.

The paper also has many grammatical inconsistencies and mistakes. These would be easy to correct, but should be done to improve the readability and flow of the paper.

---

### Decision · Action_Editor_v4uT · 2024-01-30

**Recommendation:** Accept with minor revision

**Comment:**

The paper has been reviewed by 3 reviewers: two recommend "leaning accept" (qWBR and mYsr) and one recommends "learning reject" (reviewer pViX). It provides a lower bound o the size of DeepONets, mainly as a function of the number N of training data. Some numerical results have also been provided to verify experimentally the necessary growth of output as a function of N.

During the rebuttal phase, the paper was significantly improved but the authors erroneously did not make the answers visible to reviewer pViX. This has been corrected and there has been a lively discussion to clarify the remaining issues this reviewer had.

I recommend acceptance of the paper subject to the final comments of reviewer pViX being taken into account.

**Audience:**

There is a growing literature on deep operator networks so a part of the TMLR's audience will be interested in its findings.

**Claims And Evidence:**

The claims made in the submission are accurate and convincing.

---

> ### Author Response · Authors · 2024-01-31
> **Thanks!**
>
> Thanks a lot for your comments!
>
> As you can see the version we have uploaded most recently (today) already accounts for the last comment we got from pViX.
>
> Is there anything else that we need to change?
>
> Or can we now upload a de-anonymized version in the "accepted" template of TMLR?

---

> > ### Comment · Editors_In_Chief · 2024-01-31
> > **You can go ahead an submit your camera ready**
> >
> > The AE will then be notified that they need to verify the camera ready version (e.g. if the changes made are sufficient and if the TMLR format is respected), and once they approve the submitted camera ready version the paper will be considered as published.